## TECHNIQUE

# Decoding firings of a large population of human motor units from high-density surface electromyogram in response to transcranial magnetic stimulation

Jakob Škarabot[1] , Claudia Ammann[2,3], Thomas G. Balshaw[1], Matjaž Divjak[4] , Filip Urh[4], Nina Murks[4] , Guglielmo Foffani[2,3,5] and Aleš Holobar[4]

[1] *School of Sport, Exercise and Health Sciences, Loughborough University, Loughborough, UK*
[2] *HM CINAC (Centro Integral de Neurociencias Abarca Campal), Hospital Universitario HM Puerta del Sur, HM Hospitales, Madrid, Spain*
[3] *CIBERNED, Instituto de Salud Carlos III, Madrid, Spain*
[4] *Systems Software Laboratory, Faculty of Electrical Engineering and Computer Science, University of Maribor, Maribor, Slovenia*
[5] *Hospital Nacional de Parapléjicos, Toledo, Spain*

Handling Editors: Richard Carson & Madeleine Lowery

The peer review history is available in the Supporting Information section of this article (https://doi.org/10.1113/JP284043#support-information-section).

*The Journal of Physiology*

**Abstract** We describe a novel application of methodology for high-density surface electromyography (HDsEMG) decomposition to identify motor unit (MU) firings in response to transcranial magnetic stimulation (TMS). The method is based on the MU filter estimation from HDsEMG decomposition with convolution kernel compensation during voluntary isometric

**Jakob Škarabot** is a Lecturer in Neuromuscular Physiology at the School of Sport, Exercise and Health Sciences, Loughborough University (UK). He obtained his BSc degree in kinesiology from University of Ljubljana (Slovenia), MSc degree in biomechanics from University of Jyväskylä (Finland), and PhD in neurophysiology from Northumbria University (UK). His research focuses on motor neuron physiology in health and disease. **Aleš Holobar** obtained his PhD degree in computer science from University of Maribor (Slovenia). He is a full Professor and Head of the Institute of Computer Science at the Faculty of Electrical Engineering and Computer Science, University of Maribor. His research focuses on biomedical signal processing, human–machine interfaces and rehabilitation engineering.

contractions and its application to contractions elicited by TMS. First, we simulated synthetic HDsEMG signals during voluntary contractions followed by simulated motor evoked potentials (MEPs) recruiting an increasing proportion of the motor pool. The estimation of MU filters from voluntary contractions and their application to elicited contractions resulted in high (>90%) precision and sensitivity of MU firings during MEPs. Subsequently, we conducted three experiments in humans. From HDsEMG recordings in first dorsal interosseous and tibialis anterior muscles, we demonstrated an increase in the number of identified MUs during MEPs evoked with increasing stimulation intensity, low variability in the MU firing latency and a proportion of MEP energy accounted for by decomposition similar to voluntary contractions. A negative relationship between the MU recruitment threshold and the number of identified MU firings was exhibited during the MEP recruitment curve, suggesting orderly MU recruitment. During isometric dorsiflexion we also showed a negative association between voluntary MU firing rate and the number of firings of the identified MUs during MEPs, suggesting a decrease in the probability of MU firing during MEPs with increased background MU firing rate. We demonstrate accurate identification of a large population of MU firings in a broad recruitment range in response to TMS via non-invasive HDsEMG recordings.

(Received 29 October 2022; accepted after revision 17 March 2023; first published online 23 March 2023)

**Corresponding authors** J. Škarabot: School of Sport, Exercise and Health Sciences, Loughborough University, Loughborough, UK.    Email: J.Skarabot@lboro.ac.uk

A. Holobar: Systems Software Laboratory, Faculty of Electrical Engineering and Computer Science, University of Maribor, Maribor, Slovenia.    Email: ales.holobar@um.si

**Abstract figure legend** Decomposition of high-density electromyography with blind source separation allows non-invasive identification of motor unit firings during voluntary contractions. Here we present a technique for identifying motor unit firings during elicited contractions, specifically in response to transcranial magnetic stimulation. The technique is based on estimation of motor unit filters from voluntary contractions and their application to elicited contractions. After providing proof-of-concept with simulations, we show the feasibility of the technique to identify motor unit firings and ultimately, motor unit action potentials, underpinning motor evoked potentials in the first dorsal interosseous and tibialis anterior muscles in response to single-pulse transcranial magnetic stimulation.

## Key points

- Transcranial magnetic stimulation (TMS) of the scalp produces multiple descending volleys, exciting motor pools in a diffuse manner.
- The characteristics of a motor pool response to TMS have been previously investigated with intramuscular electromyography (EMG), but this is limited in its capacity to detect many motor units (MUs) that constitute a motor evoked potential (MEP) in response to TMS.
- By simulating synthetic signals with known MU firing patterns, and recording high-density EMG signals from two human muscles, we show the feasibility of identifying firings of many MUs that comprise a MEP.
- We demonstrate the identification of firings of a large population of MUs in the broad recruitment range, up to maximal MEP amplitude, with fewer required stimuli compared to intramuscular EMG recordings.
- The methodology demonstrates an emerging possibility to study responses to TMS on a level of individual MUs in a non-invasive manner.

## Introduction

The corticospinal tract is the primary conduit of movement control signals in humans (Lemon, 2008). Investigation of the excitability of the corticospinal tract is possible by assessing responses to transcranial magnetic stimulation (TMS; Barker et al., 1985), which produces a short-lasting magnetic pulse that penetrates the low-impeding skull and generates an electric field in the brain, thus depolarising the neurons in the motor

cortex (Di Lazzaro et al., 2018). When applied to motor cortex such a stimulus evokes an electrical potential and a twitch in the target muscle. The electrical potential can be recorded with electromyography (EMG) and is known as the motor evoked potential (MEP). The size of the MEP reflects the excitability of the corticospinal tract and is sensitive to neurodegenerative disease (e.g. Ammann, Dileone et al., 2020), pharmacological agents (for review, see Ziemann et al., 2015), plasticity protocols (Ammann et al., 2017; Dileone et al., 2018; Huang et al., 2005; Nitsche & Paulus, 2000; Oliviero et al., 2011; Stefan et al., 2000) and exercise (for review, see Weavil & Amann, 2018), among others.

Whilst MEPs can be easily recorded using surface EMG, such a recording does not necessarily reveal its complexity. Indeed, TMS depolarises cortical neurons in a diffuse manner, with the subsequent electrical response the result of inputs from both excitatory and inhibitory neurons converging to simultaneously excite many motoneuron pools (Bawa & Lemon, 1993). It has been suggested that differential activation of corticospinal volleys may be studied by assessing motor unit (MU) responses to TMS (Sakai et al., 1997; Terao et al., 2000, 2001). Investigation of single MU responses to TMS has typically involved the use of intramuscular EMG (Bawa & Lemon, 1993; Brouwer & Ashby, 1990, 1992). However, the high selectivity of needle electrodes allows identification of only a small number of MUs (typically 1−2), which, whilst allowing accurate identification and analysis of MU action potentials (MUAPs; Perry et al., 1981), are not necessarily representative of the behaviour of the entire motor pool. Furthermore, due to signals becoming highly inter-ferential during greater activation levels, intramuscular EMG recordings are typically limited to identification of lower-threshold MUs (Merletti et al., 2008; Yavuz et al., 2015), which means the contribution of MUs constituting responses to higher-intensity TMS (e.g. maximal MEP response) is likely to be undetected or incomplete.

In recent decades, advances in technology and decomposition algorithms (Farina et al., 2010; Gazzoni et al., 2004; Holobar & Zazula, 2007; Kleine et al., 2007; Rau & Disselhorst-Klug, 1997) have allowed for accurate identification of firings of a relatively large population of individual MUs during voluntary contractions using high-density surface EMG recordings (HDsEMG; Holobar et al., 2010, 2014; Hu et al., 2013). The use of this methodology has greatly advanced our understanding of motor control in health (Farina et al., 2016) and disease (Gallego, Dideriksen, Holobar, Ibáñez, Glaser et al., 2015; Gallego, Dideriksen, Holobar, Ibáñez, Pons et al. 2015; Puttaraksa et al., 2019). Nevertheless, HDsEMG decomposition has largely been limited to voluntary contractions, whereas less is known about MU firings underpinning elicited contractions, such as in response to TMS. The nature of elicited contractions represents a computing challenge due to highly synchronised and superimposed MU firings. Promising results have been shown previously whereby MU firings could be identified in induced reflexes (Kalc et al., 2022a; Yavuz et al., 2015), and a limited number of MU firings were identified in response to maximal percutaneous nerve stimulation (i.e. M wave; Kalc et al., 2022b). However, responses to TMS represent their own challenges, namely the greater complexity of the multi-volley compared to single-volley responses such as H-reflexes and M-waves (Bawa & Lemon, 1993; Brouwer & Ashby, 1990, 1992). Thus, the challenge to identify the firings of individual MUs during MEPs from HDsEMG recordings remains unmet.

In this study, we aimed to ascertain the validity and accuracy of identifying firings of a large population of individual MUs that constitute an evoked motor response to single pulse TMS. We utilised the principles of the convolution kernel composition (CKC) method of HDsEMG decomposition that we recently developed for identification of MU firings underpinning the H-reflex (Kalc et al., 2022a) and M-wave (Kalc et al., 2022b); here we adapted and applied this methodology to more complex evoked responses to TMS. We relied on MU filters, a unique property of HDsEMG decomposition (Francic & Holobar, 2021), obtained from a range of voluntary contractions to estimate firing instances of many MUs during MEPs. Using synthetic signals with known MU firings we demonstrate the high precision and sensitivity of the methodological approach. Thereafter, using experimental HDsEMG signals obtained from two human muscles with strong corticomotoneuronal projections (first dorsal interosseus, FDI; and tibialis anterior, TA; Brouwer & Ashby, 1992; Palmer & Ashby, 1992), we show the feasibility of decoding firings of a large population of MUs in response to TMS of the human motor cortex in a resting and voluntarily contracting muscle.

## Methods

We first present the conceptual and theoretical framework underpinning the proposed methodology, followed by a description of the simulations and the experimental work that was undertaken to test the validity of the methodology. Lastly, we detail the approach to data analysis.

### HDsEMG model and motor unit filters

To identify MUs from multichannel surface electro-myogram, we used the principles of the CKC algorithm (Holobar & Zazula, 2007). The reader is referred to the previously published material for further details on the mathematical approach to the CKC method

(Holobar & Zazula, 2007; Holobar et al., 2014). Briefly, HDsEMG signals were modelled as the multiple input–output system described previously (Holobar & Zazula, 2007):

$$x_i = \sum_{j=1}^{N} \sum_{l=0}^{L-1} h_{ij}(l) t_j(n-l); i = 1, \ldots, M \qquad (1)$$

where $h_{ij}$ is a MUAP of $L$ samples of the $j$-th MU, as detected by the $i$-th electrode of HDsEMG with $M$ channels.

Blind source separation algorithms, such as the CKC method, invert the EMG mixing model and estimate the so-called MU filters (Fig.1). The MU filter is a unique, MU specific set of weights in the linear spatio-temporal combination of the HDsEMG channels that yields the estimation of the individual MU spike train (Francic & Holobar, 2021). Here, the MU spike train is defined as:

$$t_j(n) = \sum_{k} \delta \left(n - \tau_j(k)\right), \ j = 1 \ldots N \qquad (2)$$

Where $\delta$ denotes the unit-sample pulse, and $\tau_j(k)$ denotes the time of the $k$-th MU firing. In other words, $t_j(n)$ consists of zeros and ones, whereat ones denote the time moments of MU firings.

The MU filter is then defined as (Francic & Holobar, 2021):

$$\text{MU fllter} = \sum_{P} Y^T \left(n_p\right) C_y^{-1} \qquad (3)$$

Where $Y(n) = [y_1(n), y_1(n-1) \ldots y_1(n-F), y_2(n) \ldots y_M(n-F)]^T$ represents a MF $\times$ 1 vector of multichannel EMG signals, extended by factor $F = 15$ or similar (Holobar & Zazula, 2007), $C_y^{-1}$ is the inverse of the MF $\times$ MF correlation matrix of $Y$, and $n_p$ denotes the firing time of a given MU in a voluntary contraction.

The conceptual framework of MU filters is depicted in Fig.1. The classic spatial filters applied to HDsEMG recordings (e.g. monopolar, Laplacian) may increase its spatial selectivity, but the selectivity in the spatial domain is insufficient to discriminate a given MU spike train from noise (i.e. crosstalk from another MU; Fig.1*B*). Thus, an increase in the number of weights in both the spatial and the temporal domain is required for a given MU filter to minimise the crosstalk from other MUs, allowing threshold-based segmentation of MU spike trains to discriminate MU firings from baseline noise (Fig.1*C*). Notably, a MU filter can be estimated in a short-portion of a HDsEMG recording (e.g. 10 s) and applied to the remaining portion of the recording from the same muscle performing the same contraction (Glaser et al., 2013). Furthermore, in isometric conditions, an MU filter can be estimated from recordings during one contraction and applied to recordings of another contraction performed by the same muscle (Francic & Holobar, 2021). The efficiency of MU filter estimation on one and its application to another contraction has been recently demonstrated on both synthetic and experimental HDsEMG signals (Francic & Holobar, 2021). An iterative estimation and application of filters in successive, short time windows has also been shown to be sufficiently robust to identify MU firings under conditions that would otherwise render successful decomposition of the entire signal unlikely due to MUAP changes as a result of changes in muscle fibre length and conduction velocity, such as during prolonged, sustained contractions (Rossato et al., 2022) and during dynamic contractions (Kramberger & Holobar, 2021), respectively.

In the present study, we tested and validated a methodology that extends the concept of estimation and re-application of MU filters from isometric voluntary contractions to contractions elicited by TMS of the motor cortex. With this approach we are relying on the inherent strengths of the blind source separation algorithms such as the CKC method, namely: (1) rather than their application, the difficulty is the establishment of MU filters, which, in this study, are obtained from extensively validated procedures applied to HDsEMG recordings of voluntary isometric contractions (Holobar et al., 2010, 2014); and (2) the inherent ability to resolve highly complex superimpositions of MUAPs from HDsEMG once the initial MU separation filter is established (Glaser et al., 2013).

Furthermore, one also needs to consider some physiological factors that likely facilitate the success of the described methodology. First, voluntary contractions and contractions elicited by TMS likely share the recruitment order (Bawa & Lemon, 1993), meaning that the MU separation filters identified during the former are likely to be applicable to the latter. Second, the differences in excitability and conduction velocity of pyramidal tract and $\alpha$-motoneurons, and the differences in innervation zone location and dispersion result in a dispersion of MU firings that is sufficient for the identification of true MU firings even in highly synchronised MUAPs constituting elicited contractions with TMS.

## Experiment 1. Simulation on synthetic signals

To assess the efficiency of MU filter transfer from voluntary to elicited contractions, we generated synthetic EMG signals with known MU firings. Synthetic HDsEMG signals were generated by a multilayer cylindrical volume conductor model (Farina et al., 2004). The model consisted of 40 mm-thick bone, 30 mm-thick muscle, 4 mm-thick fat and a 1 mm-thick skin layer (Holobar et al., 2010). The simulation involved 10 fusiform muscles with an elliptical cross-sectional area, with the depth and width

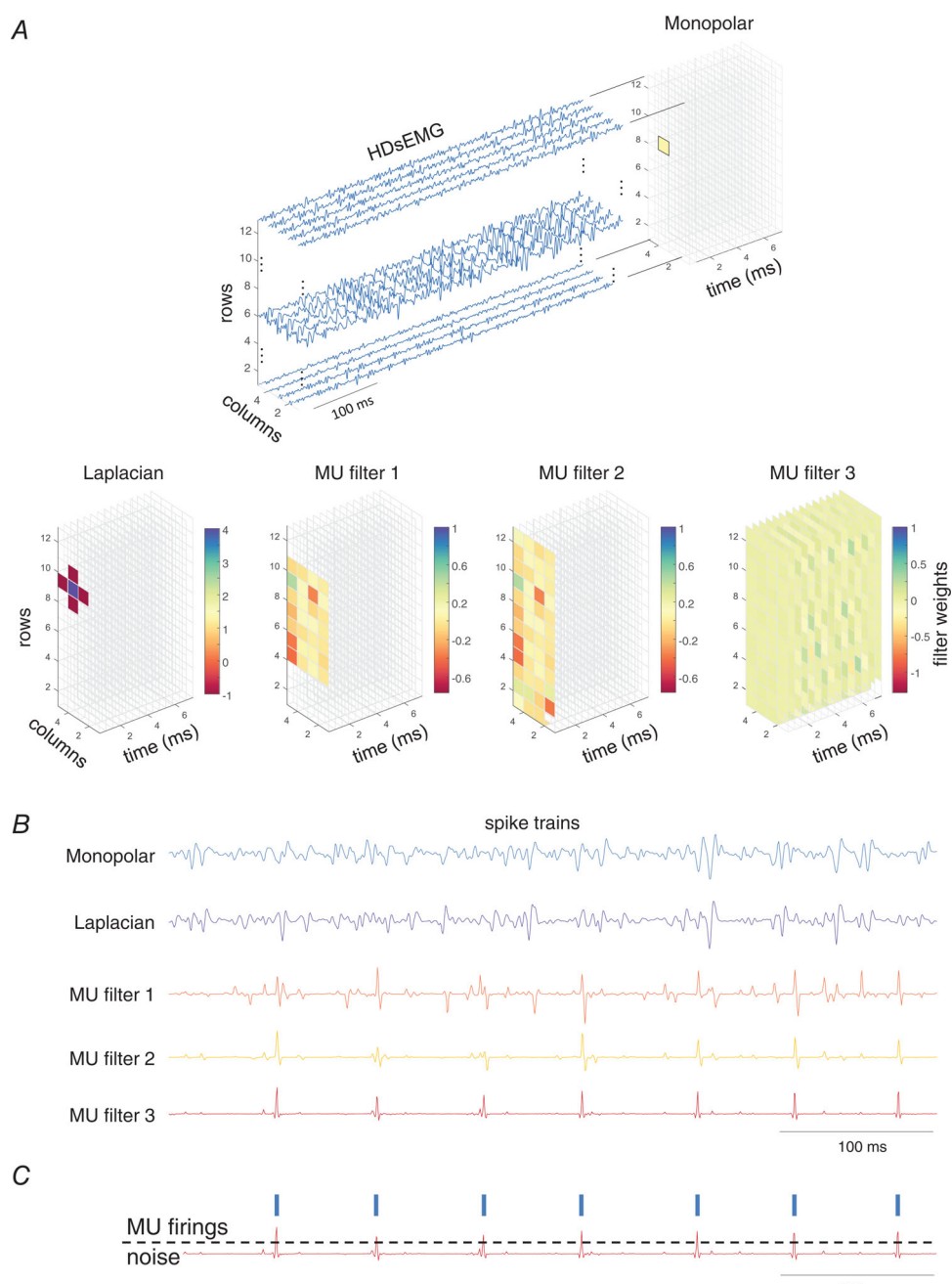

**Figure 1. The concept of motor unit (MU) filters and MU spike trains**

*A*, different filters applied to high-density electromyography (HDsEMG) signals, recorded by an array of 5 × 13 surface electrodes from the first dorsal interosseous muscle during a constant isometric contraction at 20% of maximal voluntary force. Classical spatial filters (monopolar, Laplacian) and MU specific filters that are estimated by a blind source separation algorithm (MU filters 1, 2 and 3) are depicted. The filter weights are presented by coloured rectangles with the colour bar scaled in the range of minimal and maximal values of weights for individual filters (note that for MU filter 3, the weights with the higher coefficients may be obstructed by layers of lower filter weights). *B*, increasing the number of weights of MU filters decreases the noise (crosstalk from other MUs) in estimated spike trains. In a convolutive mixing model of HDsEMG (see Eqn 1 in text) the MU filter weights can extend in both spatial and temporal direction (MU filter 3), supporting effective suppression of MU crosstalk in an estimated MU spike train. *C*, MU firings are estimated by threshold-based spike segmentation, separating the estimated MU spikes (denoted by blue vertical bars) from baseline noise. [Colour figure can be viewed at wileyonlinelibrary.com]

radius set to 15 and 30 mm, respectively, an average fibre length of 130 mm, the spread of those fibre endings of 5 mm, and the innervation zone centred longitudinally across the muscle fibres (Keenan et al., 2005). For each muscle, 200 MUs (Keenan et al., 2005) were randomly and uniformly distributed within the muscle tissue with sizes ranging from 24 to 2408 fibres (Kukulka & Clamann, 1981), and the size and recruitment threshold properties following Henneman's size principle of many small and fewer progressively bigger MUs (Henneman, 1957). Further, the conduction velocity of MUs was set to have a normal distribution with the mean value of 4.0 (0.3) m/s (Keenan et al., 2005), with all fibres belonging to a given MU having the same conduction velocity.

A set of progressively increasing contractions of 10 s in duration was simulated for each muscle: 10, 30, 50, 70 and 90% of maximal voluntary force (MVF). These contractions resulted in 105, 155, 178, 193 and 200 recruited MUs, respectively. The MU firing patterns were generated using the Fuglevand et al. (1993) model adopted for the biceps brachii muscle, with the recruitment range set to 0–80% MVF, a linear increase in firing rate from 8 to 35 spikes/s at recruitment and during MVF, respectively, and the coefficient of variation for the interspike interval set to 20% for each MU.

The simulated voluntary contractions were followed by sets of 97 MEPs simulated with an increasing proportion of the recruited motor pool (10, 25, 50, 75, 100%); this simulated the MEP recruitment curve in response to stimulations with increasing intensity. For each simulated MEP, a normal distribution of firing latencies was set to 10 ms with a standard deviation of 1.3 ms (note that the crucial parameter for MU synchronization is the standard deviation, whereas the latency does not impact the assessment of the proposed methodology – see section 'Extraction of motor unit statistics'). Such a level of synchronisation is expected to approximate the physiological range of MU firing during evoked potentials (Palmer & Ashby, 1992). For each simulated MEP, we removed one-third of randomly selected MU firings. This removal facilitated unique identification of a simulated MU that was used as a reference when calculating the sensitivity and precision (see Eqn 6). Namely, without this step, simulated MUs would have shared too many firings (pairwise comparison with a tolerance of MU firing match set to ±0.5 ms) to support accurate recognition of which MU was identified from simulated MEPs, especially in cases of less accurate MU identification.

Lastly, HDsEMG signals were simulated with an electrode array of 90 electrodes (10 × 9; 5 mm inter-electrode distance, 0.5 mm electrode diameter). To simulate individual HDsEMG channels, the channel specific MUAPs were convolved with the simulated MU firing pattern for each HDsEMG channel, and the resulting MUAP trains from all the MUs were summed.

The simulated HDsEMG had an added coloured Gaussian noise with a bandwidth of 20−500 Hz and signal-to-noise ratio set to 25 dB. To replicate the conditions of experimental HDsEMG recordings, the synthetic signals were generated at a sampling rate of 4096 Hz, then down-sampled to 2048 Hz; thus, the simulated MU firings also occurred between individual HDsEMG samples.

### Experiment 2. Experimental recordings of responses to TMS in first dorsal interosseous muscle

**Participants and ethical approval.** The measurements of FDI were conducted at HM CINAC, Hospital Universitario HM Puerta del Sur, Móstoles, Madrid, Spain. Seven adults (2 females; age: 35 (7) years, stature: 1.73 (0.62) m, mass: 67.9 (10.8) kg) without a history of neurological or psychiatric conditions, and who were not taking medication, volunteered to participate in the experiments. Written informed consent was given by participants prior to experimental procedures. The study was approved by the Comité Ético de Investigación de HM Hospitales (15.03.764-GHM) and conducted in accordance with *Declaration of Helsinki*, except for registration in database.

**Experimental protocol and procedures.** Participants were seated on a comfortable chair and were instructed to keep their eyes open for the duration of the experiment and to relax the left arm and hand on a pillow while the right arm was resting on a table. The right arm was fixed in a custom-made isometric brace, supporting the isometric abduction of index finger and immobilization of other fingers. The force of index finger abduction was measured by universal digital force gauges (Sauter FH 100, Sauter GmbH, Germany; measuring frequency: 2000 Hz, resolution: 0.05 N. The skin above FDI was cleaned with ethanol and abraded (EVERI160SPE, Spes Medica, Italy). A disposable, single-channel surface bipolar EMG sensor (biEMG) was placed on the FDI. The surface EMG signals were amplified and band-pass filtered (×1000, 2−2000 Hz; D360, Digitimer Ltd, Welwyn Garden City, UK), sampled at 5 kHz (Power 1401; Cambridge Electronic Design (CED) Ltd, Cambridge, UK), and acquired using Signal software (version 5; CED).

Following biEMG electrode placement, hotspot and resting motor threshold (RMT) were determined. Mono-phasic single-pulse transcranial magnetic stimuli were delivered via a 70-mm figure-of-eight coil connected to a Magstim 200² stimulator (Magstim Co. Ltd, Whitland, UK). The coil was positioned tangential to the scalp, with the handle oriented backwards and at 45° with respect to the midline, to induce a posterior–anterior cortical current in the motor cortex. RMT was defined as the minimum TMS output intensity evoking a MEP

amplitude of 50 $\mu$V in 5 out of 10 trials. After that, the single-channel EMG sensor was replaced by a multichannel, high-density surface EMG (HDsEMG) grid consisting of 64 electrodes (13 rows × 5 columns, 1 mm electrode diameter, 4 mm inter-electrode distance; GR04MM1305, OT Bioelettronica, Torino, Italy), covered by disposable bi-adhesive foam layer (Spes Medica, Battipaglia, Italy) the cavities of which were filled with conductive paste (Ten20, Weaver and Co., Aurora, CO, USA). A strap electrode dampened with water was placed on the contralateral wrist. The HDsEMG signals were recorded in monopolar configuration, bandpass filtered (10–500 Hz), digitised (2048 Hz) via a multichannel amplifier (16-bit, Quattrocento; OT Bioelettronica), and acquired using OT Biolab+ software (OT Bioelettronica).

A warm-up was then performed, consisting of 3−5 s submaximal isometric index finger abductions (3 × 50, 3 × 75, and 1 × 90% of perceived maximum). After 120 s of rest, two MVF measurements were performed with verbal encouragement provided to the participant and 120 s of rest between the measurements. Following determination of MVF, participants performed sustained isometric index finger abductions of 15−20 s in duration at 10, 20, 30, 50 and 70% MVF. A rest period of 120 s was provided after each contraction. After the performance of voluntary contractions, single pulse TMS was delivered to the left primary motor cortex area (FDI hotspot) to induce responses in the right resting FDI muscle. TMS output intensities ranged between 80 and 140% of the resting motor threshold (RMT) in 10% increments (randomised order). Twenty stimuli were delivered per stimulation intensity (a total of 140 stimuli). During the experiment, stimuli were delivered every 6 (1) s, consistent with previous experiments (Ammann, Guida et al., 2020). Stimulation triggers were delivered to OTBiolab+ via auxiliary inputs of a multichannel amplifier (Quattrocento) for synchronisation with HDsEMG signals.

After TMS stimulation, participants performed another set of sustained isometric index finger abductions of 15−20 s in duration at 10−70% MVF. These post-TMS voluntary contractions were used for the assessment of MU filter reliability.

## Experiment 3. Experimental recordings of responses to TMS in tibialis anterior muscle

**Participants and ethical approval.** Nine adults (2 females; age: 26 (4) years, stature: 1.76 (0.11) m, mass: 75.1 (16.6) kg) were recruited from Loughborough University and participated in the experiments. Participants had no known neurological disorders or musculoskeletal injury limiting contractions with the lower limb, were not taking any medication known to affect the nervous system, and reported no contraindication to TMS (Keel et al., 2001). The experimental procedures were approved by Loughborough University Ethical Committee (2021-5011-4261) and were conducted in accordance with the *Declaration of Helsinki* except for registration in database. Participants provided written, informed consent prior to taking part in experiments.

**Experimental protocol and procedures.** Each experimental session started with preparation of skin for EMG placement involving shaving, abrasion and cleaning with ethanol. A single-channel surface biEMG was then placed over the muscle belly of TA (fixed 1-cm inter-electrode distance; Trigno Standard EMG sensors, Delsys, Boston, MA, USA), approximately 1 cm lateral to the tibial crest (Vieira et al., 2017). The surface EMG signals were amplified and band-pass filtered at source (×300; 20−450 Hz), sampled at 5 kHz (Micro 1401-3; CED), and acquired using Spike2 software (version 10; CED). During the experiment, participants were seated on a massage table upon which an ankle ergometer fitted with a calibrated force transducer (CCT Transducers s.a.s., Torino, Italy) was tightly secured. The ankle was positioned at 100° of plantar flexion (90° = anatomical position) with the hip and knee at 110 and 180°, respectively. All testing was performed on the dominant limb. The foot was secured on the ergometer foot plate with two straps (35-mm wide reinforced canvas webbing) on the distal third of metatarsals and foot dorsum. The analog signal from the transducer was amplified (×200) and sampled at 5 kHz for synchronisation with surface EMG (Micro 1401-3; CED) and HDsEMG (Quattrocento; acquired with OTBiolab+ software; OT Bioelettronica).

Single-pulse stimuli were delivered using a concave double-cone coil connected to a magnetic stimulator (Magstim 200$^2$). The coil was positioned over the presumed leg area of the primary motor cortex contralateral to the target limb and oriented to induce posterior-to-anterior cortical current. A hotspot in the TA was determined as described previously (Škarabot, Ansdell, Brownstein, Hicks et al., 2019) and was marked directly on the scalp using indelible ink to ensure consistent coil placement throughout the trial. RMT was then defined as the intensity evoking an EMG response of 50 $\mu$V in 5 out of 10 trials. All stimuli were triggered using customised scripts in Spike2 (v10; CED).

After that, the single-channel bipolar surface EMG sensor was replaced by a HDsEMG grid (GR08MM1305) over the muscle belly of TA via a disposable bi-adhesive foam layer. The cavities of the foam layer were filled with conductive paste. A reference electrode (Kendall Medi-Trace, Cardinal Health, Dublin, OH, USA) was placed over the medial malleolus. A strap electrode dampened with water was

placed on the ankle of the contralateral leg to ground the electrodes. The HDsEMG signals were recorded in monopolar mode, bandpass filtered (10–500 Hz), digitised (5120 Hz) via a 16-bit multichannel amplifier (Quattrocento), and acquired using OT Biolab+ software

A warm-up was performed consisting of 3—5 s submaximal isometric dorsiflexions (3 × 50, 3 × 75 and 1 × 90% of perceived maximum), followed by two to three isometric dorsiflexions with maximal effort to determine their MVF. Participants then performed a series of submaximal isometric trapezoidal contractions with dorsiflexors at 10, 20, 30, 50, 70 and 90% MVF, slowly increasing/decreasing the force level to the target in 3, 4, 5, 6, 7 and 8 s, respectively, and maintaining constant torque at the target for 20 (10–30% MVF), 15 (50% MVF), 10 (70% MVF) or 5 s (90% MVF). A single trapezoidal contraction was performed for force levels between 10 and 50% MVF, whereas two and three trials were performed during contractions at 70 and 90% MVF, respectively. Not all participants were capable of sustaining force level at 90% MVF during trapezoidal contractions; in such cases, those trials were discarded. The order of trapezoidal contractions was randomised, except for those at 90% MVT, which were always performed last to minimise fatigability.

Following trapezoidal contractions and 5 min of rest, participants received single-pulse TMS during rest. Eight pulses were delivered at each stimulus intensity, ranging from 90 to 180% RMT (or till plateau in MEP response, or the maximal stimulator intensity, whichever occurred sooner) in 10% RMT increments, totalling a maximum of 80 stimuli. Stimuli were delivered 6—7 s apart in a block of eight, with the order of stimulation intensities randomised. Stimulation triggers were simultaneously delivered via auxiliary inputs of a multichannel amplifier for synchronisation with HDsEMG signals. Relative to Experiment 2, we used a lower number of stimuli and a greater range of stimulation intensities, with the aim of a principal focus on stimulation intensity to assess the feasibility of decomposing MU firings associated with a maximal MEP response. Finally, to allow assessment of reliability of HDsEMG decomposition, participants repeated the same series of submaximal isometric trapezoidal contractions with dorsiflexors (10–90% MVT) as at the beginning of the experiment.

### Experiment 4. Motor unit firings in response to TMS in tibialis anterior muscle during voluntary contraction

**Participants and ethical approval.** Six healthy adults from the population of Loughborough University volunteered for the experiment (1 female; age: 27 (5)

years, stature: 1.79 (0.10) m, mass: 78.2 (16.5) kg). The inclusion/exclusion criteria were the same as for Experiment 3. The experimental procedures were approved by Loughborough University Ethical Committee (2021-5011-4261) and were conducted in accordance with the *Declaration of Helsinki* except for registration in database. Participants provided written, informed consent prior to taking part in experiments.

**Experimental protocol and procedures.** The experimental procedures relating to biEMG, HDsEMG, force recordings and TMS were the same as for Experiment 3. Initially, a biEMG electrode was placed on the muscle belly of TA. Participants then performed a series of isometric dorsiflexion contractions, including a standardised warm-up and contractions with maximal effort to determine MVF. After that, hotspot, active motor threshold (AMT) and the intensity corresponding to the maximal MEP response were determined. AMT was defined as the stimulus intensity corresponding to MEP amplitude of 200 $\mu$V in 3 out of 5 trials whilst participants were isometrically contracting the dorsiflexors at 10% MVF. Following determination of hotspot and stimulation intensities, biEMG sensors were replaced with a HDsEMG electrode grid. A series of isometric trapezoidal contractions was then performed (10–90% MVF, similar to Experiment 3). After that, a contraction–response curve was constructed with stimuli delivered whilst participants were contracting at 10, 30, 50, 70 and 90% MVF (30–90 s rest between contractions). Stimuli were delivered at 120% AMT. Five stimuli were delivered at each contraction strength. Finally, the same series of voluntary contractions (10–90% MVF) as before the stimulations was repeated at the end to allow the assessment of reliability of MU filters.

### Data analysis

**Bipolar EMG recordings.** Peak-to-peak amplitude of MEPs obtained with biEMG recordings were analysed live for the purpose of determination of hotspot and RMT using customised scripts in Signal and Spike2 software, respectively. No off-line analyses were performed on the biEMG signals.

**Decomposition of high-density electromyography.** Both synthetic and experimental signals were decomposed in monopolar configuration using the CKC method (Fig.2*A*; Holobar & Zazula, 2007). Each voluntary contraction was decomposed independently (e.g. Fig.2*D*) and the quality of the decomposition was determined by a previously validated metric of pulse-to-noise ratio (PNR), which allows for a computationally efficient estimation of MU crosstalk in a CKC-based estimation of the MU spike train

(Holobar et al., 2014). The PNR is defined as (Holobar et al., 2014):

$$\text{PNR} \ \left(t_j(n)\right) = 10 \times \log\left(\frac{E\left(t_j^2(n)\,|_{t_j(n)=1}\right)}{E\left(t_j^2(n)\,|_{t_j(n)=0}\right)}\right) \quad (4)$$

where $E$ represents mathematical expectation, $t_j(n)|_{t_j(n)=1}$ is the segmented MU spike denoting MU firing and $t_j(n)|_{t_j(n)=0}$ denotes the baseline noise (MU crosstalk). The reader is directed towards previously published material for further information on the full mathematical approach to calculation of PNR (Holobar et al., 2014).

Following the recommendations in Holobar et al. (2014), all MUs exhibiting PNR < 28 dB (sensitivity of MU firing identification <85%) were discarded. The remaining MU firing patterns were visually inspected, segmented into MU firings using in-built functions of the CKC method, and edited by an experienced expert operator with an approach described previously (Del Vecchio et al., 2020). Briefly, the MU filter was iteratively optimised by removing the portions of spike trains of poorer quality, followed by recalculation of the MU filter as per Eqn 3 (Del Vecchio et al., 2020). Additionally,

MUs with a firing pattern exhibiting abnormal firing rates (interspike interval <20 ms or >250 ms) and/or high level of irregularity (coefficient of variation of interspike interval >0.4) were discarded. To identify unique MUs across different contraction levels, the decomposed and edited MU spike trains of individual voluntary contractions were concatenated (Fig.2*B*). The estimated MU filter was then recalculated using the segmented firing moments $n_p$ of a given MU and applied to concatenated HDsEMG signals $Y(n)$ to estimate the MU spike train of a given unit $t(n)$ that was recruited across all the contraction levels:

$$t(n) = \sum_p Y^T(n_p)\,C_y^{-1}Y(n) \quad (5)$$

For each identified MU, we repeated the segmentation of MU spikes $t(n)$ into MU firings $n_p$ and baseline noise (Fig.2*C*). Afterwards, the MU firing patterns $n_p$ were mutually compared, and duplicates removed by discarding the MU with the smallest PNR. The duplicates were considered those that shared at least 30% of the same MU firings, with firing match tolerance set to 0.5 ms. For each remaining MU, we recalculated PNR metrics using different temporal supports in Eqn 4. In other words, we searched for the longest epoch of the MU spike

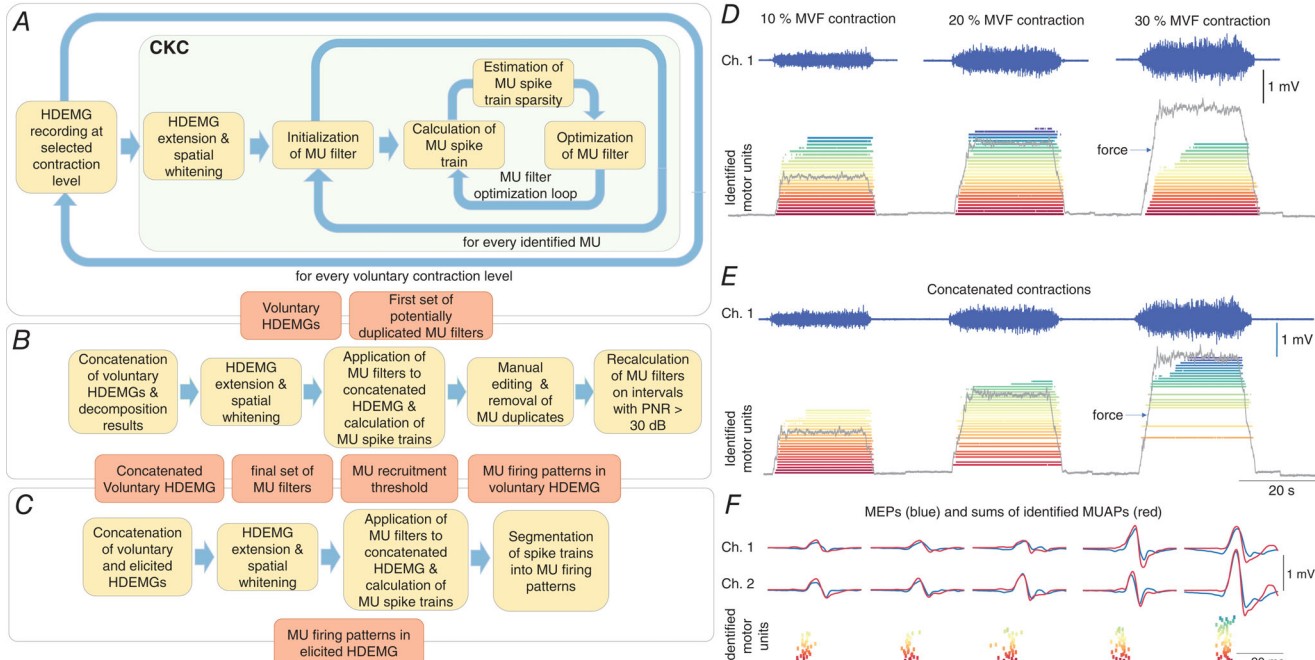

**Figure 2. A convolutive pipeline for identification of motor unit (MU) firings during motor evoked potentials (MEPs)**
*A*, identification of MU filters and MU firing patterns from high-density surface electromyography (HDsEMG) signals recorded during individual voluntary contraction levels. *B*, concatenation of voluntary HDsEMGs and removal of MU duplicates. *C*, application of MU filters to elicited HDsEMGs. *D*, examples of recorded HDsEMGs during individual voluntary contractions and corresponding MU identifications. *E*, an example of concatenated voluntary HDsEMGs and sets of unique MU firing patterns with temporal support limited to regions with a pulse-to-noise ratios >30 dB. *F*, examples of the identified MU firings in MEPs (for clarity, only two channels are shown). [Colour figure can be viewed at wileyonlinelibrary.com]

train $t_j(n)$ that exhibited PNR > 30 dB (for example, see Fig.2*E*), suggesting accuracy >90% and false alarm rate <5% (Holobar et al., 2014). Finally, we used only the MU firing moments $n_p$ identified from this epoch of the MU spike train $t_j(n)$ to recalculate the MU filter again, as per Eqn 5. This approach ensured the highest possible quality of MU filters extracted from concatenated HDsEMG signals of voluntary contractions.

Following concatenation of HDsEMG signals obtained from voluntary contractions and reliable calculation of MU filters of unique MU spike trains, concatenation of voluntary and elicited contractions was performed. For elicited contractions, stimulation artifacts were removed using an artifact blanking procedure with a 5 ms window from the stimulation trigger. MU filters were then applied from voluntary to elicited contractions (Fig.2*C*; for example, see Fig.2*F*). Therefore, MU filters were calculated and optimised only on voluntary signals, with no additional optimisation of MU filters performed once they were applied to elicited contractions. To identify the MU spike trains in elicited contractions, individual MU spikes were segmented into MU firing and baseline noise using the threshold-based spike segmentation (Fig.1*C*) of the CKC method (Holobar & Zazula, 2007). The segmentation of the identified spike trains in elicited contractions was additionally manually inspected by an expert operator and corrected, if necessary. The same procedure as proposed for voluntary contractions (Del Vecchio et al., 2020) was followed, with extra attention paid to PNR value of the identified MU spike train $t_j(n)$ in the local vicinity of MEP (Holobar & Zazula, 2007; Holobar et al., 2009, 2010, 2014) where we compared the energy of spikes to the energy of baseline noise to identify and correct potential problems with spike segmentation into MU firings and in MU crosstalk in the identified MU spike train during MEPs. It is noteworthy that these corrections included MU spike train segmentation only, whereas MU filters and MU spike trains $t_j(n)$ remained unchanged.

A similar approach was performed when assessing reliability of MU filters throughout experiments, i.e. the voluntary contractions performed before and after all the stimulations were concatenated, and the MU filters calculated from voluntary contractions performed prior to stimulations were applied to voluntary contractions performed at the end of the experiment (without applying them to MEP recordings).

It is noteworthy that the CKC algorithm identifies MU spike trains with individual spikes corresponding to the close proximity of a MUAP peak in the HDsEMG signal (Holobar & Zazula, 2007), resulting in an inherent and consistent delay of all the spikes of a given MU in the order of a few milliseconds. Considering differences in MUAP shapes between different MUs, the inherent delays will thus differ among a given pool of identified

MUs, resulting in methodologically induced dispersion of identified MU firing latencies. To compensate for this methodological artefact, we first estimated MUAPs of identified MUs using spike-trigger averaging of HDsEMG signals from voluntary contractions with MU firings $n_p$ used as triggers. After that, we plotted the multichannel MUAPs along with the relative position of the MU spikes used as triggers (custom-made MATLAB (The MathWorks, Natick, MA, USA) script). For each MU, we then manually estimated the MU spike delay with respect to the earliest detected start of the MUAP across HDsEMG channels (i.e. the earliest deflection of the MUAP waveform from baseline) and subtracted this delay from the identified MU latencies in the MEP.

**Extraction of motor unit statistics**

*Synthetic signals.* To assess decomposition accuracy in synthetic signals, the identified MU firings were compared to simulated firings, with precision and sensitivity of MU firings identification calculated as follows (Francic & Holobar, 2021):

$$\text{Precision} = \frac{\text{TP}}{\text{TP} + \text{FP}}, \text{Sensitivity} = \frac{\text{TP}}{\text{TP} + \text{FN}} \quad (6)$$

with TP, FP, and FN denoting the number of true positive, false positive and false negative firings, respectively, with the tolerance limit of MU firing match set to 0.5 ms. Additionally, we calculated the PNRs of the MUs identified during the simulated voluntary and elicited contractions.

*Experimental signals.* From voluntary contractions, we calculated the recruitment threshold of all identified MUs and their PNR. For voluntary contractions with superimposed MEPs, we also calculated the firing rate of MUs in the 1000 ms preceding the stimulus. Regarding global HDsEMG amplitude, peak-to-peak amplitude was calculated from the interference signal, considering only the channel that consistently exhibited the largest response size in the electrode grid. When considering MEPs, we extracted the number of uniquely identified MUs in response to TMS. To assess representativeness of MU identification in MEPs, firing patterns of the identified MUs were convolved to yield a MUAP train. The representativeness was then quantified in HDsEMG signals by calculating the energy accounted for (EAF) in the signal by decomposition (Holobar et al., 2010):

$$\text{EAF (\%)} = \left( \frac{E\left[ y_i(n) - \sum_j m_{ij}(n)^2 \right]}{E\left[ y_i^2(n) \right]} \right) \times 100 \quad (7)$$

where $E$ represents mathematical expectation, $y_i(n)$ denotes the $i$-th HDsEMG channel and $m_{ij}(n)$ denotes the MUAP train of the $j$-th MU in the $i$-th HDsEMG

channel. Here, consideration was given to the previously demonstrated decomposition bias towards identification of MUs close to the uptake electrode, with distant MUs constituting physiological noise (Divjak et al., 2020; Holobar et al., 2009). Thus, to remove the contribution of distant and unidentifiable MUs to HDsEMG signals and provide a better representative of the yield when decomposing MEPs, a 2D Laplacian (LP) spatial filter was applied to the MUAPs and MEPs when calculating energy accounted for by MEP decomposition. The energy accounted for in the signal by decomposition was calculated for both voluntary and elicited contractions to allow comparison.

Lastly, we quantified the variability in the identification of firings of individual MUs to assess the consistency in MU firing identification. Specifically, we calculated the mean standard deviation of the latency of firings of individual MUs (SD$_{\text{MULat}}$). When assessing latencies of MU firings during MEPs, one needs to consider the methodological factors that will likely contribute to the measured variability of latencies at the detection site. Due to highly synchronised evoked responses and thus synchronised MUAPs, small differences in MUAP conduction velocities and the length of single muscle fibres might result in the dispersion of firing detection delays (Fig.3). Furthermore, differences in the dispersion of innervation zones and end plate locations will likely cause further dispersion in the detection of firing latencies of individual MUs. These methodological factors are not unique to the proposed methodology and have been identified previously in intramuscular signals (Bawa et al., 2002), but bear noting when making interpretations about the observed behaviour of the firing latencies of identified MUs during MEPs. To quantify this dispersion, we additionally calculated the standard deviation of firing latencies of all the identified MUs in a given MEP (SD$_{\text{MULat,MEP}}$).

### Statistical analysis

Statistical analyses were performed in R (R Studio v 1.4.1106, R Foundation for Statistical Computing, Vienna, Austria). To take into account the dependence and different number of observations (number of MUs, responses to stimulation with the same stimulation intensity; Giboin et al., 2020; Yu et al., 2022), linear mixed models were constructed with stimulation intensity as fixed effect, and participant as random intercept (*lme4* package; Bates et al., 2015). For Experiment 4, contraction level replaced stimulation intensity as fixed effect in the model. The modelling was performed separately for each outcome variable. Normality of distribution was assessed with histograms and quantile–quantile plots of residuals. For outcomes with skewed data distribution, we used generalised instead of linear mixed models. For

analyses of MEP amplitude and the number of identified MUs, all stimulation intensities were considered. Subsequent analysis, however, only involved stimulation intensities above the motor threshold ($\geq$110% RMT). The significance of models was assessed with *lmerTest* package (Kuznetsova et al., 2017). For outcomes related to synthetic signals (precision, sensitivity), the model was constructed such that it considered all voluntary contraction levels from which MU filters were estimated individually or together. Pairwise *post hoc* tests of estimated marginal means were performed with the Tukey adjustment for multiple comparisons (*emmeans* package; Lenth & Lenth, 2018). To estimate the recruitment order of MU firings in MEPs, the repeated measures correlation coefficient was computed for the association between

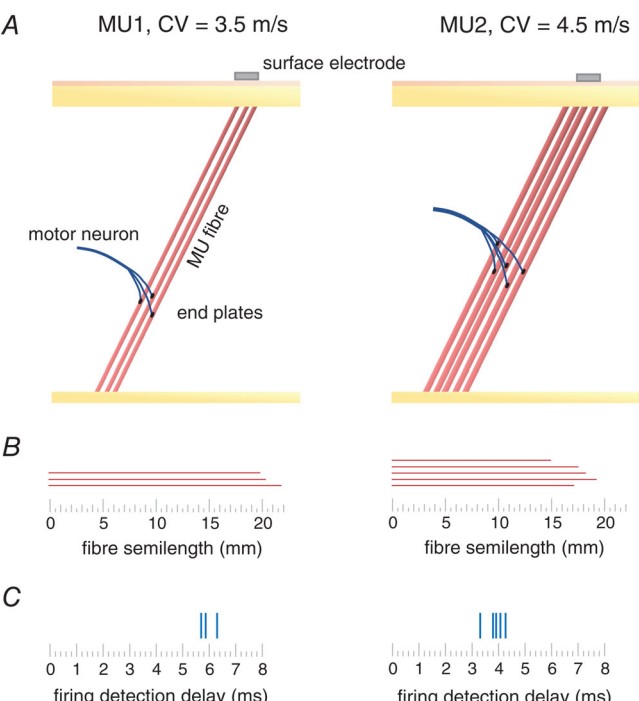

**Figure 3. Delay between single fibre depolarisation at the neuromuscular junction and the detection of motor unit (MU) firing at the electrode**

*A*, schematic representation of low recruitment threshold motor unit (MU 1) with conduction velocity (CV) of 3.5 m/s and semi-fibre length of ∼21 mm and high recruitment threshold MU (MU 2) with CV of 4.5 m/s and semi-fibre length of ∼17 mm. For reasons of clarity only a few MU fibres are depicted. Surface EMG electrode is depicted by grey rectangle. *B*, semi-fibre lengths of fibres depicted in *A* as measured from end plates to the superficial end of the fibres. *C*, delays of single fibre action potential (SFAP) detection by the depicted surface electrode, under the assumption that, when measured at the surface of the skin, SFAPs of the depicted MU consist mostly of end of fibre effects and that all the firing delays were created due to propagation of SFAP along the muscle fibre (no effect of motoneuron conduction velocity is considered). At the depicted semi-fibre lengths and CVs, the activity of MU 1 is detected with delay of ∼5.5 ms, whereas the activity of MU 2 is detected with delay of ∼3.4 ms. [Colour figure can be viewed at wileyonlinelibrary.com]

**Table 1. The number, recruitment thresholds and accuracy (pulse-to-noise ratio, PNR) of the identified motor units (MUs) during voluntary contractions and contraction evoked with transcranial magnetic stimulation**

| | Experiment 1 (*n* = 10, simulated biceps brachii) | Experiment 2 (*n* = 7, first dorsal interosseous) | Experiment 3 (*n* = 9, tibialis anterior) | Experiment 4 (*n* = 6, tibialis anterior) |
|---|---|---|---|---|
| **Voluntary contractions** | | | | |
| Total sample of MUs | 1370 | 264 | 406 | 297 |
| MU number per *n* | 18.0 (3.0) | 37.7 (9.5) | 45.1 (15.4) | 49.5 (10.7) |
| Range of MUs per *n* | 14–20 | 28–51 | 16–61 | 30–60 |
| Mean MU recruitment threshold (%MVF) | 31.6 (21.0) | 20.7 (6.7) | 15.4 (4.7) | 11.8 (4.5) |
| MU recruitment threshold range (%MVF) | 2.0–80.0 | 0.8–59.0 | 0.2–63.9 | 0.1–46.6 |
| PNR start of experiment (dB) | 35.4 (5.3) | 36.3 (4.1)*** | 33.8 (3.0)*** | 33.6 (2.7)*** |
| PNR end of experiment (dB) | — | 34.7 (4.5) | 31.7 (3.8) | 31.9 (3.2) |
| EAF (monopolar) (%) | 35.6 (7.6) | 32.0 (4.5) | 12.1 (11.6) | 9.8 (11.7) |
| EAF (LP-filtered) (%) | 51.6 (9.2) | 34.5 (5.6) | 29.4 (15.2) | 24.1 (34.0) |
| **Evoked contractions** | | | | |
| Total sample of MUs | 746 | 215 | 365 | 272 |
| MU number per *n* | 8.8 (0.5) | 30.7 (9.9) | 28.9 (17.1) | 45.3 (11.6) |
| Range of MUs per *n* | 7–10 | 12–40 | 8–59 | 30–60 |
| Mean MU recruitment threshold (%MVF) | 27.7 (19.4) | 15.9 (6.3) | 8.2 (5.1) | 12.2 (3.9) |
| MU recruitment threshold range (%MVF) | 2.0–80.0 | 0.8–57.3 | 0.2–63.3 | 0.1–46.6 |
| PNR (dB) | 36.7 (3.5) | 34.9 (5.5) | 31.6 (4.7) | 31.5 (4.4) |

Where appropriate, data are presented as mean (SD). *n* represents simulated muscles for Experiment 1 and participants for Experiments 2–4.
*** $P < 0.001$ compared to the end of experiment. MVF, maximal voluntary force; EAF, energy accounted for.

the recruitment threshold of MUs during voluntary contraction and the number of MU firings detected in MEPs across the entire recruitment curve (*rmcorr* package; Bakdash & Marusich, 2017). Similarly, a repeated measures correlation coefficient was calculated for MEP amplitude and the number of identified MUs during a MEP. To describe the relationship between MU firing rate and the probability of evoked response, we relied on the fact that we were able to extract firings of many MUs per individual evoked response. This is in contrast to a previously reported approach to the calculation of the probability of evoked response with intramuscular signals, which relies on a great number of evoked responses due to the limited number of MU firings detected in an individual response to TMS (Bawa & Lemon, 1993). Here, a firing of a given MU was classified as a binary variable, and a mixed effect logistic binomial regression was performed with the background MU firing rate as fixed effect, and each MU and participant taken as random

intercepts (*lme4* package). From there, the probability of MU firing during a MEP was estimated by calculating the marginal effects on the response scale (*ggeffects* package; Lüdecke, 2018). Significance was set at an $\alpha$ level of 0.05. Data are presented as means (SD), except for correlation coefficients where data are presented as means (95% confidence interval).

## Results

### Experiment 1. Simulation on synthetic signals

An example of 200 simulated MUs, four synthetic HDsEMG signals and the identified spike from five MUs are shown in Fig.4A. Data on the decomposition yield for synthetic signals are presented in Table 1.

The simulation on synthetic signals demonstrated that transferring MU filters from voluntary contractions to MEPs recruiting a progressively increasing proportion of the motor pool, and thus simulating the recruitment

curve, resulted in very high precision and sensitivity of MU identification during simulated MEPs (>90% on average), though they varied across stimulation intensities and were dependent on voluntary contraction levels from which MU filters were estimated (Fig.4*B*

and *C*). When considering MU filters obtained from all contraction levels, precision of MU firing identification during MEPs decreased with stimulation intensity ($\chi^2$ (4) = 39.9, $P < 0.0001$). With 100% stimulation intensity, precision was lower (95.3 (1.2)%) compared to

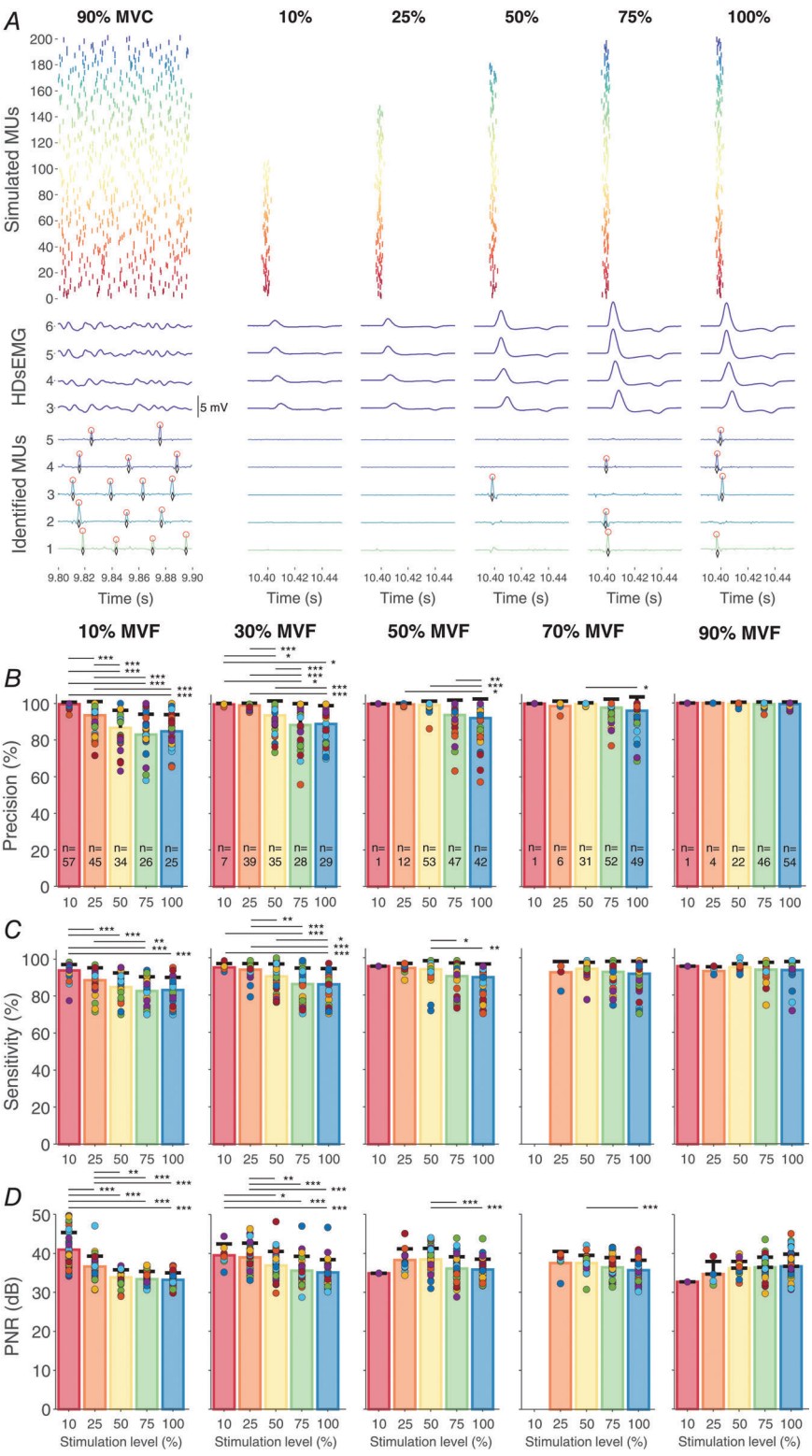

**Figure 4. The identification of firings of individual motor units (MUs) during modelled motor evoked potentials (MEPs) of varying sizes**

*A*, simulated firing patterns of individual MUs (top panel), high-density surface electromyography (HDsEMG) signals (middle panel), and the identified MU spike trains (bottom panel) during voluntary contractions at 90% of maximal voluntary contraction force (MVF; left) and MEPs simulated with varying stimulation intensities (expressed as a 10, 25, 50, 75 and 100% of motor pool excitation; right). In simulated MEPs, the standard deviation of firing latency of individual MUs was set to 1.3 ms. For clarity, only 100 ms of voluntary contractions, four HDsEMG channels, and MU spike trains of five identified MUs are shown, with black diamonds denoting true firings and red circles denoting the identified MU firings. The identified MUs were of relatively high threshold and were thus only identified during large simulated MEPs. *B–D*, precision (*B*), sensitivity (*C*), and pulse-to-noise ratios (PNR; *D*) of firings of the identified MUs (denoted by coloured circles) during MEPs simulated with varying stimulation intensities. Each column represents the results of the firings of individual MUs during simulated MEPs that were identified from MU filters obtained from simulated high-density electromyography signals at different contraction levels (10, 30, 50, 70, and 90% of MVF). Data are presented as means with SD shown by bars, with individual MUs shown as circles. In *B*, the number (*n*) of the identified MUs identified across 10 simulated muscles is also displayed as bars. The horizontal lines denote the statistical difference between different stimulation intensities: *$P < 0.05$, **$P < 0.01$, ***$P < 0.001$. [Colour figure can be viewed at wileyonlinelibrary.com]

the stimulation intensity at 10% (99.3 (0.9)%, $P < 0.0001$) and 25% (95.7 (1.4)%, $P = 0.0069$). Precision was also lower at 75% stimulation intensity (95.1 (1.5)%) compared to 10% ($P < 0.0001$) and 25% ($P = 0.0179$), and at 50% (95.0 (1.7)%) compared to 10% ($P = 0.0058$). Sensitivity also decreased with stimulation intensity ($\chi^2$ (4) = 28.1, $P < 0.0001$). *Post hoc* testing indicated that sensitivity was greater at 10% (94.0 (1.6)%) compared to 50% (90.7 (1.3)%, $P = 0.0438$), 75% (90.7 (1.2)%, $P = 0.0001$) and 100% stimulation intensity (91.0 (1.1)%, $P < 0.0001$), but not 25% (90.8 (1.6)%, $P = 0.0719$). Similar results were obtained when considering filters acquired during individual voluntary contraction levels, with the highest precision and sensitivity displayed for MUs identified during MEPs simulated with lower compared to higher stimulation intensities (Supporting information, 'Statistical analysis of precision, sensitivity and PNR in simulated MEPs'; Fig.4*B* and *C*). Though fewer MUs were identified at higher than at lower contraction levels, both precision ($\chi^2$ (4) = 327.3, $P < 0.0001$) and sensitivity ($\chi^2$ (4) = 209.9, $P < 0.0001$) of MU firing identification were greater for MU filters acquired during higher compared to lower contraction levels (Fig.4 and *C*).

The average PNR for the identified MUs during simulated MEPs was 36.7 (3.5) dB, though this was affected by stimulation intensity ($\chi^2$ (4) = 38.3, $P < 0.0001$), such that it was greater during 10% compared to all higher intensities ($P < 0.0001$), during 25% compared to 75% and 100% ($P < 0.0001$), and during 50% compared to 75% ($P = 0.0270$) and 100% ($P = 0.0020$). However, at all stimulation intensities the PNR values were above 30 dB, suggesting accurate identification of spikes of individual MUs in response to TMS (Holobar et al., 2014). Similar results were found when considering transferring of MU filters from individual contraction levels (Supporting information 'Statistical analysis of precision, sensitivity and PNR in simulated MEPs'; Fig.4*D*).

### Experiment 2. Decomposition of motor evoked potentials in the first dorsal interosseus

The number of identified MUs during voluntary and evoked contractions, along with metrics of decomposition accuracy are presented in Table 1. The identified MU firings in a typical participant during MEPs elicited at several stimulation intensities are depicted in Fig.5*A*. Peak-to-peak MEP amplitude increased with stimulation intensity ($\chi^2$ (6) = 262.0, $P < 0.0001$; Fig.5*B*). Consistent with the notion of increases in the amplitude of the evoked responses, the number of identified MUs during MEPs increased with greater stimulation intensity ($\chi^2$ (6) = 977.3, $P < 0.0001$), from 7.7 (5.2) MUs for MEPs

elicited at 110% RMT to 13.9 (7.5) MUs during MEPs elicited at 140% RMT ($P \leq 0.0106$; Fig.5*C*). A positive association was demonstrated between the number of identified MU firings and MEP amplitude ($r_{rm}$ = 0.65 (0.62, 0.69), $P < 0.0001$), meaning that with greater MEP amplitude, more MU firings were identified per MEP. No differences were detected for the number of identified MU firings when responses were elicited with stimulation intensities of 80, 90, and 100% of RMT (0.1 (0.1) MU firings on average; $P = 0.9993$–1.0000).

The identified MUs accounted for 31.2 (12.0)% of LP-filtered MEP energy, though this was influenced by stimulation intensity ($\chi^2$ (3) = 19.8, $P = 0.0002$), with the accounted energy being lower for identified MUs during stimulation at 110% RMT compared to 130% ($P = 0.0002$) and 140% RMT ($P = 0.0028$; Fig.5*D*). There was a negative correlation between the number of firings of the identified MUs during MEPs and the MU recruitment threshold estimated during voluntary contractions ($r_{rm}$ = −0.82 (−0.86, −0.77), $P < 0.0001$; Fig.6*A*), suggesting that, similar to voluntary contractions, the recruitment of MUs in response to TMS was orderly.

Examples of post-stimulus time histograms of several identified MUs in FDI for one participant are shown in Fig.7*A*. On average, the latency of the identified MUs during MEPs was 23.1 (1.3) ms; however, this was dependent on stimulation intensity ($\chi^2$ (3) = 21.3, $P < 0.0001$). The firing latency of the identified MUs was found to be longer during MEPs elicited at 140% RMT compared to lower stimulation intensities ($P \leq 0.0008$; Fig.5*E*). Notably, however, these differences were in the order of 0.1–0.2 ms. Considering the sampling rate of HDsEMG recordings of FDI was set to 2048 Hz, the observed increase in MU firing latency is below one-half of the intersample interval of HDsEMG (0.5 ms), and thus the observed differences in the firing latency of the identified MUs are negligible. For the latency at the level of individual MUs, low variability was demonstrated ($SD_{MULat}$ 0.6 (0.3) ms) suggesting consistency of firing identification of individual MUs. The standard deviation ($\chi^2$ (3) = 65.9, $P < 0.0001$; Fig.5*F*) of latencies of all the identified MUs in a given MEP, indicating dispersion of MU firing identification, increased with greater stimulation intensity ($SD_{MULat,MEP}$: $P \leq 0.0068$), likely as a result of the recruitment of higher threshold MUs.

### Experiment 3. Decomposition of motor evoked potentials in tibialis anterior

Data on the number of identified TA MUs during voluntary and evoked contractions are shown in Table 1. An example of MU firings identified during MEPs elicited at several stimulation intensities is depicted in Fig.8*A*. Peak-to-peak amplitude of MEP increased

with greater stimulation intensity ($\chi^2$ (9) = 653.6, $P < 0.0001$; Fig.8B), though a decrease was observed at 180% RMT. Note, however, that due to having reached 100% of stimulator output, the number of participants at higher stimulation intensities is lower compared to lower stimulation intensities, which could explain this apparent decrease. Consequently, the identified number of MUs in

response to TMS increased with stimulation intensity ($\chi^2$ (9) = 263.6, $P < 0.0001$; Fig.8C), meaning that additional MUs were recruited with greater magnetic stimulus to the scalp, though again the number of identified MUs decreased at 180% RMT. A positive association was found between the peak-to-peak MEP amplitude and the number of identified MU firings ($r_{rm} = 0.53$

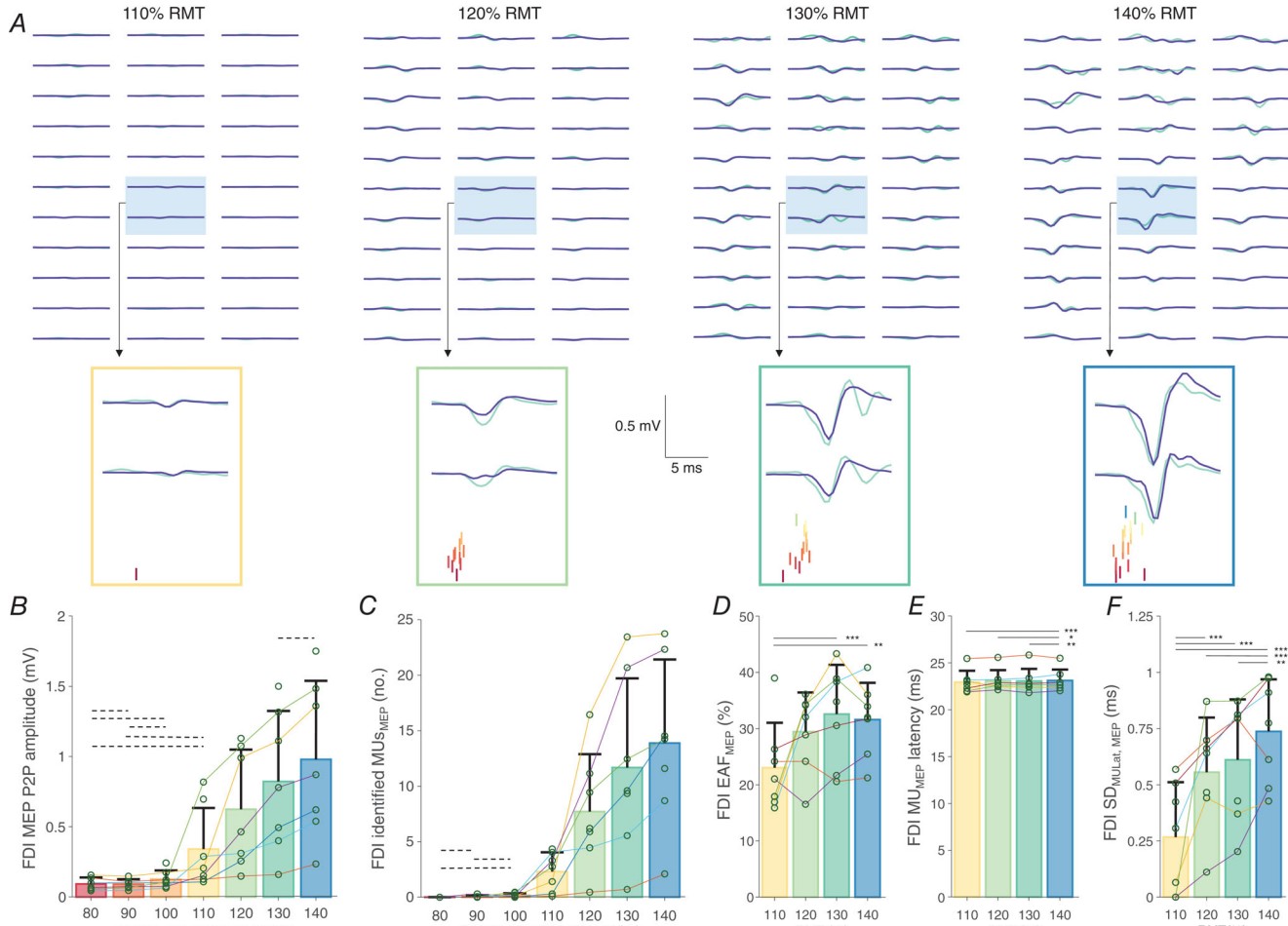

**Figure 5. Identification of firings of a population of motor units (MUs) in first dorsal interosseous (FDI) during the motor evoked potential (MEP) recruitment curve**
*A*, examples of typical MEPs in FDI in response to transcranial magnetic stimulation (TMS) at 110−140% of resting motor threshold (RMT). High density surface electromyography (HDsEMG) channels (green line) filtered with a Laplacian spatial filter (LP) are displayed along with their sum of spike-trigger averaged MU action potentials (blue line). Two channels are highlighted along with the corresponding identified MU firings (coloured). Note that responses and thus the proportion of the reconstructed waveform with respect to the recorded MEPs may vary across the channels due to factors such as distribution of MU territories and (mis)alignment of MUAPs with respect to MEP. The average proportion of accounted energy in LP-filtered MEPs by the identified MU firings in the two highlighted channels was 20% (110% RMT), 33% (120% RMT), 48% (130% RMT) and 49% (140% RMT). *B*, maximal peak-to-peak (P2P) amplitude of MEPs across LP-filtered HDsEMG channels in response to TMS between 80 and 140% RMT. *C*, the number of identified MUs during MEPs in response to TMS between 80 and 140% RMT. *D*, the proportion of accounted energy in LP-filtered MEPs by the identified MU firings (energy accounted for, EAF). *E*, the latency of the identified MU firings during MEPs. *F*, the standard deviation of latencies of all the identified MU firings during MEPs (SD$_{MULat,MEP}$), indicating dispersion of MU firing identification. For *B–F*, data are presented as mean with SD shown as bars, with individual participant averages shown as circles. The dashed horizontal lines in *B* and *C* denote non-significant differences between responses to different stimulation intensities. In *D–G*, the statistical difference between different stimulation levels is denoted by full horizontal lines: *$P < 0.05$, **$P < 0.01$, ***$P < 0.001$. [Colour figure can be viewed at wileyonlinelibrary.com]

(0.47, 0.58), $P < 0.0001$). The number of identified MU firings did not differ for MEPs elicited at 90 and 100% RMT (3.4 (6.0) MU firings on average; $P = 0.0565$), but then a general increase of the identified MUs was observed for MEPs evoked with up to 160% RMT (17.5 (17.9) MUs), followed by a relative decrease at 180% RMT (9.1 (9.4) MU firings; Fig.8*C*), likely due to the difficulty in identifying lower-threshold MUs during very large responses consistent with simulation experiments (Experiment 1).

There was a negative correlation between the number of identified MU firings across the MEP recruitment curve and the recruitment threshold of those MUs during voluntary contractions ($r_m = -0.75$ ($-0.80$, $-0.68$), $P < 0.0001$; Fig.6*B*), suggesting the recruitment of additional MUs with increased stimulus intensity occurred in an orderly fashion. The identified MUs accounted for 26.9 (8.5)% of LP-filtered MEP energy, which was not affected by stimulation intensity ($\chi^2$ (7) = 4.5, $P = 0.7152$; Fig.7*D*).

The post-stimulus time histograms of several identified MUs in TA from an individual participant are depicted in Fig.8*B*. The latency of MU firings identified in MEPs was 36.5 (4.5) ms on average, which was dependent on stimulation intensity ($\chi^2$ (7) = 75.0, $P < 0.0001$). The latency of the identified MU firings was longer at 110% RMT compared to at >130% RMT ($P < 0.0001$), whereas the latencies were the shortest for MEPs evoked at 180% RMT compared to other stimulation intensities ($P \leq 0.0002$; Fig.8*E*). The variability of latencies at the level of individual MUs was low ($SD_{MULat}$ 2.7 (1.3) ms), suggesting relative consistency in the identification of MU firings. The dispersion of latencies within a given MEP

($SD_{MULat,MEP}$) was on average 2.9 (1.3) ms (Fig.8*F*), which was not affected by stimulation intensity ($\chi^2$ (6) = 11.0, $P = 0.1382$; $\chi^2$ (6) = 10.3, $P = 0.1704$).

## Experiment 4. Decomposition of motor evoked potentials evoked during voluntary contractions

Results on the number of identified MUs during voluntary contractions and in response to TMS are shown in Table 1. An example of the identified MU firings during elicited contractions are depicted in Fig.9*A*. The MU firing rate in the 1 s preceding the stimuli increased progressively with contraction level ($\chi^2$ (4) = 2452.3, $P < 0.0001$; Fig.9*B*). The MU firings during elicited contractions were negatively associated with background MU firing rate ($\chi^2$ (1) = 24.2, $P < 0.0001$), suggesting that the greater the MU firing rate, the lower the probability of firing of the identified MU during evoked responses (Fig.9*C*). The number of identified MUs during MEPs was less predictably influenced by the background contraction level ($\chi^2$ (4) = 21.5, $P = 0.0003$; Fig.9*E*), such that the number of identified MUs increased for MEPs evoked at 10% compared to 30% MVF ($P = 0.0306$), and then, compared to 30%, decreased at 70% ($P = 0.0225$) and 90% MVF ($P = 0.001$). The peak-to-peak amplitude of MEPs was expectedly influenced by contraction level ($\chi^2$ (4) = 18.2, $P = 0.0011$), although it was found to be only smaller at 10% MVF compared to contraction levels equal to or greater than 50% MVF ($P \leq 0.0263$; Fig.9*D*). These findings suggest that whilst the firing probability of MUs during MEPs decreased with increased background contraction level, additional, non-identified MUs

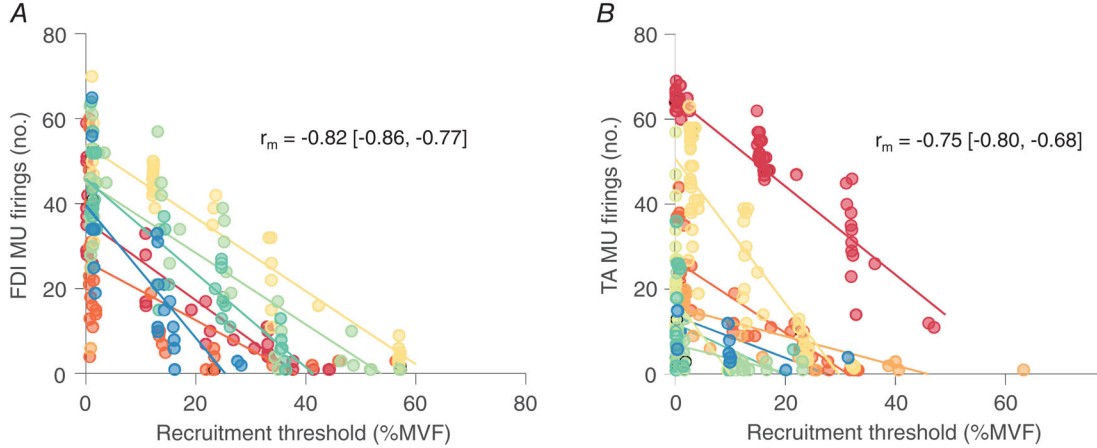

**Figure 6. The relationship between the number of MU firings identified across the MEP recruitment curve and their voluntary recruitment threshold in first dorsal interosseus (FDI; *A*) and tibialis anterior (TA; *B*)**

Circles denote the number of times a firing of a given MU was identified during MEPs across the entire recruitment curve as a function of its voluntary recruitment threshold. Each colour represents an individual participant. A significant negative repeated measures correlation coefficient ($r_m$ [95% CI]) is shown. [Colour figure can be viewed at wileyonlinelibrary.com]

were recruited during MEPs at higher contraction levels (70–90% MVF).

The identified MUs on average accounted for 24.0 (6.7)% of LP-filtered MEP energy, which was independent of background contraction level ($\chi^2$ (4) = 6.8, $P$ = 0.1463). The latency of MU firings identified in MEPs was 35.1 (3.8) ms, which was similar across the contraction levels ($\chi^2$ (4) = 7.9, $P$ = 0.0970; Fig8$G$). At the level of individual MUs, the variability of latency was relatively low (SD$_{MULat}$ 4.5 (4.4) ms; CoV$_{MULat}$ 12.6 (11.5)%), though seemingly higher than what was observed when identifying MU firings during MEPs evoked in a resting muscle (see Experiment 3). The dispersion of latencies within individual MEPs was

influenced by the background contraction level (SD$_{MEPLat}$ average: 5.4 (2.0) ms; $\chi^2$ (4) = 90.7, $P$ < 0.0001) with a tendency to increase with greater background contraction levels (Fig.8$H$).

## Discussion

We described and validated a new methodology to identify firings of a large population of MUs during MEPs in response to TMS from HDsEMG. The methodology is based on the blind source separation principles of the CKC method, which involves estimating MU filters from voluntary contractions and afterwards applying them to elicited contractions to identify MU firings comprising MEPs. We first simulated synthetic EMG signals during

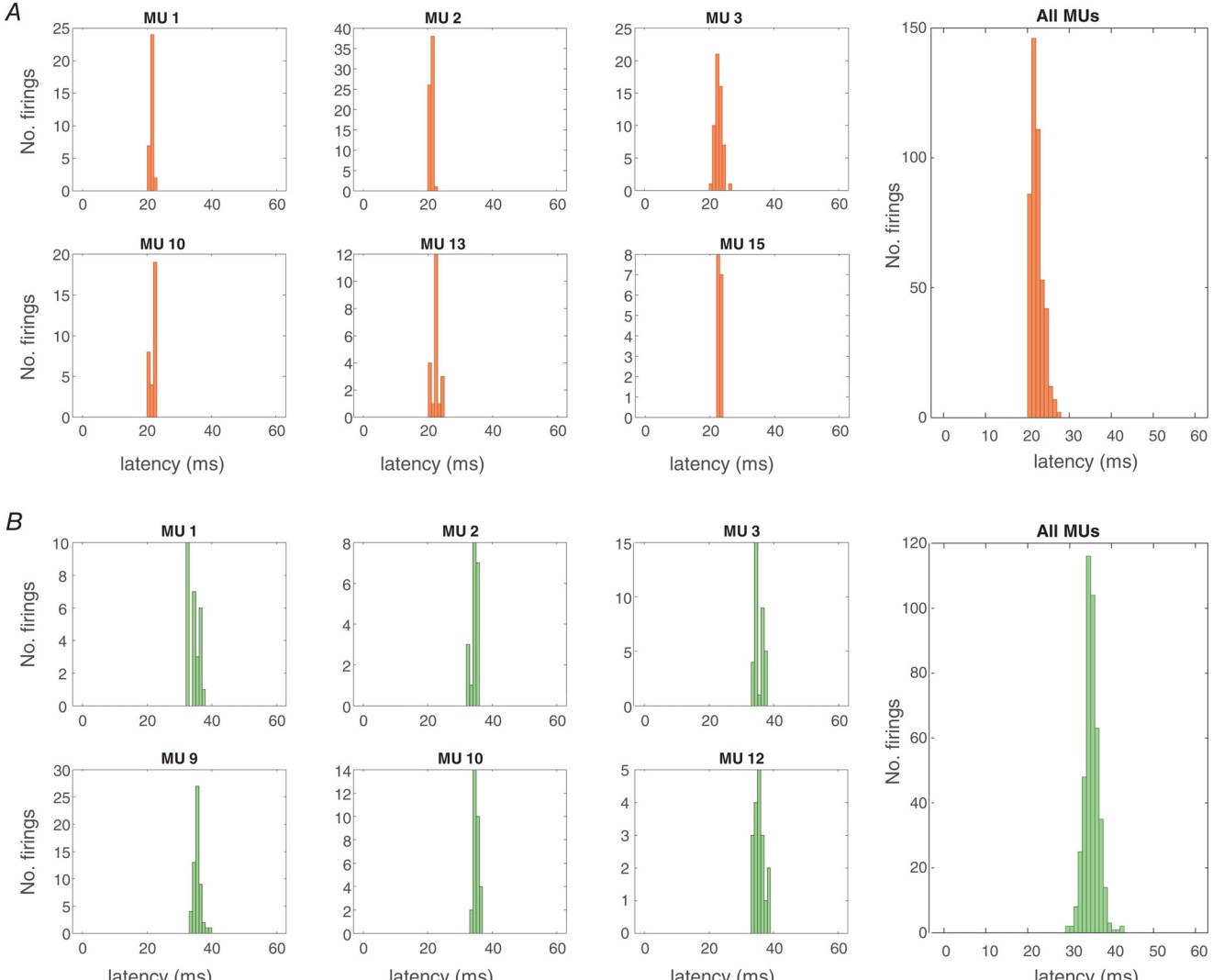

**Figure 7. Post-stimulus time histograms**
Post-stimulus time histograms (bin width: 1 ms) of firings of the identified motor units (MU) in first dorsal inter-osseous (FDI; *A*) and tibialis anterior (TA; *B*) muscles across the entire motor evoked potential recruitment curve in two individual participants. For reasons of clarity, individual histograms of 6 MUs per muscle are depicted (left small panels), along with the histogram for all identified MUs (right panel). [Colour figure can be viewed at wileyonlinelibrary.com]

different levels of voluntary and elicited contractions and showed that the described methodology displays high precision and sensitivity, with low false alarms. We then tested the feasibility of the methodology by identifying firings of a large population of MUs during the MEP recruitment curves in FDI and TA, as well as during MEPs elicited during a series of background voluntary contraction levels. We demonstrated low variability in the identified MU firing latencies and that the energy of LP-filtered MEPs accounted for by decomposition was comparable to the values typical for HDsEMG decomposition of voluntary contractions (Francic & Holobar, 2021; Holobar

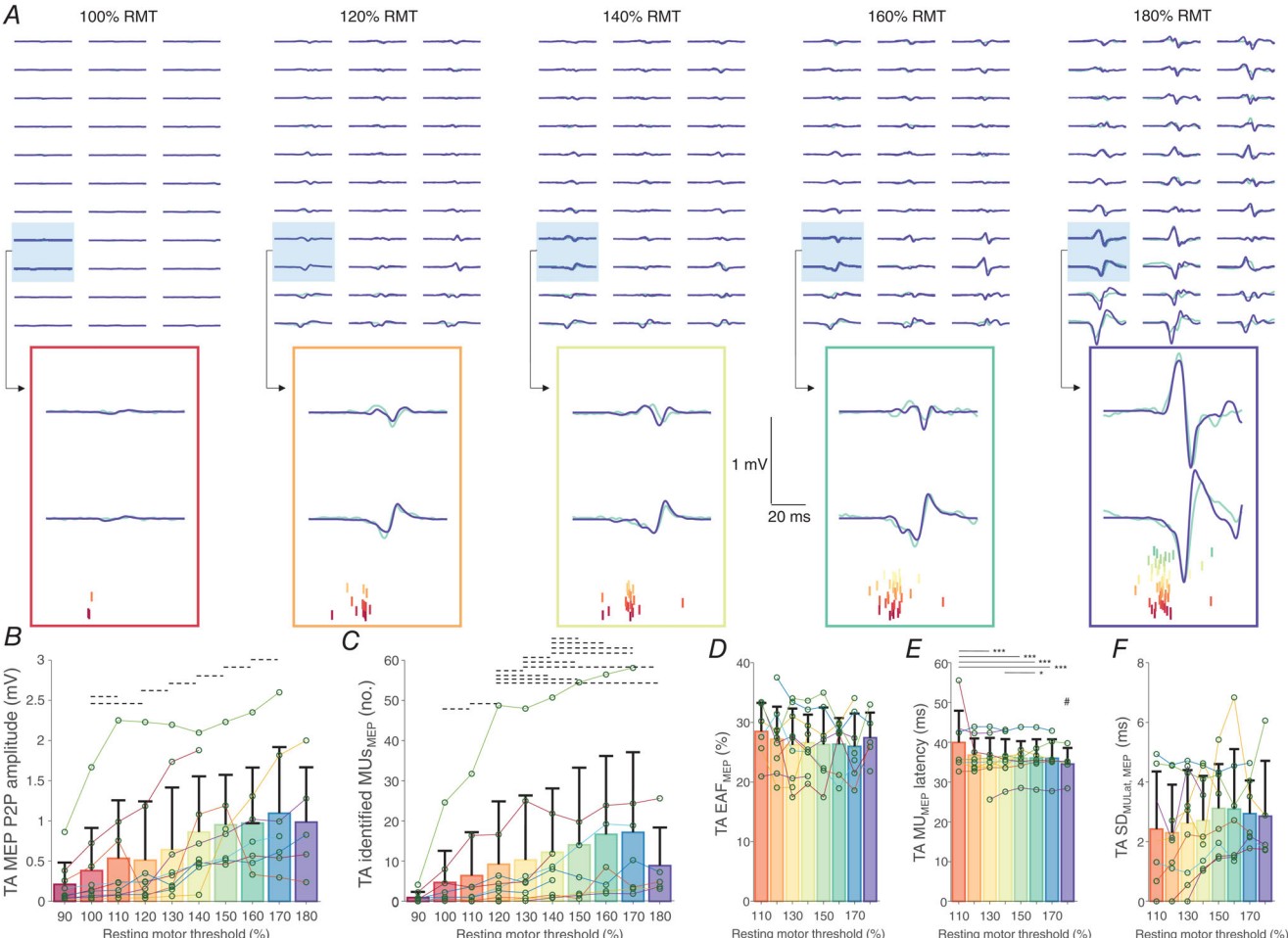

**Figure 8. Identification of firings of a population of motor units (MUs) in tibialis anterior (TA) during the motor evoked potential (MEP) recruitment curve**

*A*, examples of typical MEPs in TA, filtered with a Laplacian spatial filter (LP), in response to transcranial magnetic stimulation (TMS) at 100, 120, 140, 160 and 180% of resting motor threshold (RMT). LP-filtered high-density electromyography (HDsEMG) channels (green line) are shown along with the sum of spike-trigger averaged MU action potentials (blue line). Additionally, two highlighted channels are displayed with the identified MU firings (coloured). Note that responses and thus the proportion of the reconstructed waveform with respect to the recorded MEPs may vary across the channels due to factors such as distribution of MU territories and (mis)alignment of MUAPs with respect to MEP. The average proportion of accounted energy in LP-filtered MEPs by the identified MU firings in the two highlighted channels was 46% (100% RMT), 52% (120% RMT), 32% (140% RMT), 38% (160% RMT) and 63% (180% RMT). *B*, maximal peak-to-peak (P2P) amplitude of MEPs across LP-filtered HDsEMG channels in response to TMS between 90 and 180% RMT. *C*, the number of identified MUs during MEPs in response to TMS between 90 and 180% RMT. *D*, the proportion of accounted energy in LP-filtered MEPs by the identified MU firings (energy accounted for, EAF). *E*, the latency of the identified MU firings during MEPs; *F*, the standard deviation of latencies of all the identified MU firings during MEPs (SD$_{MULat,MEP}$), indicating dispersion of MU firing identification. For *B–F*, data are presented as means with SD shown as bars, with individual participant averages shown as circles. The dashed horizontal lines in *B* and *C* denote non-significant differences between responses to different stimulation intensities. In *E*, the statistical difference between different stimulation levels is denoted by full horizontal lines (*$P < 0.05$, ***$P < 0.001$), with # denoting the difference relative to responses to all the other stimulation intensities. [Colour figure can be viewed at wileyonlinelibrary.com]

& Farina, 2021; Holobar et al., 2014). Furthermore, we demonstrated a series of physiological insights regarding the relationship between the number of identified MUs and MEP amplitude, recruitment order of MUs by TMS, and the probability of MU firing during MEPs with increasing background contraction levels that are consistent with the findings observed with intra-muscular EMG.

## Signal-based metrics supporting the validity of the methodology

To assess the feasibility of the methodology we first simulated synthetic HDsEMG signals for which the

ground truth about MU firings was known. We previously showed in simulations that the approach of estimating MU filters from the decomposition of voluntary sub-maximal contractions and applying them to evoked responses allows accurate identification of MU firings at a fixed proportion of motor pool recruitment, even in cases of supraphysiological synchronisation of firings (Kalc et al., 2022a, 2022b). Here, we simulated MEPs corresponding to a wide range of the proportion of motor pool activation (10–100%) and demonstrated high precision and accuracy of MU identification (>90%). Furthermore, the identified MU firings displayed PNRs greater than 30 dB as recommended previously

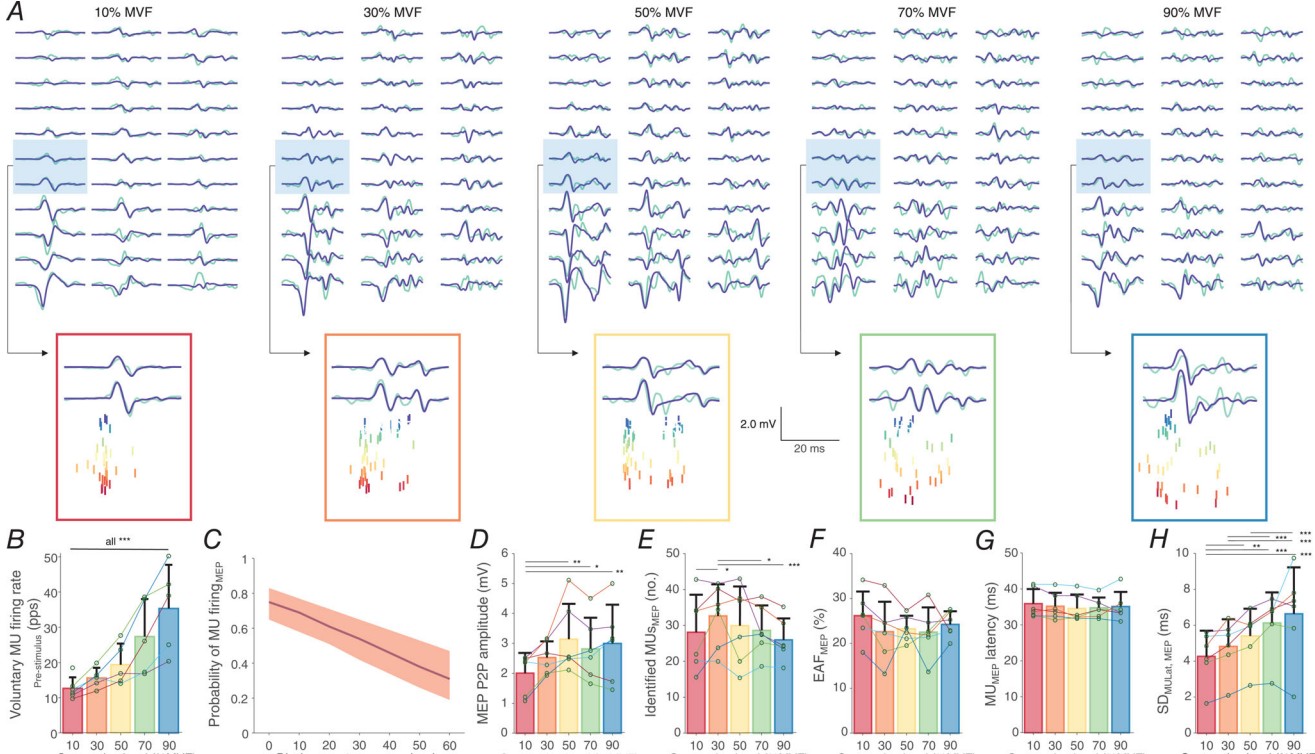

**Figure 9. Identification of firings of a population of motor units (MUs) during motor evoked potentials (MEPs) evoked during background contractions at 10, 30, 50, 70 and 90% of maximal voluntary contraction force (MVF)**
*A*, examples of typical motor evoked potentials (MEPs) filtered with a Laplacian spatial filter (LP) from a grid of high-density surface electromyography (HDsEMG) channels (green line) along with the sum of spike-trigger averaged MU action potentials (blue line). Two highlighted channels are also shown with the identified MU firings (coloured). Note that responses and thus the proportion of the reconstructed waveform with respect to the recorded MEPs may vary across the channels due to factors such as distribution of MU territories and (mis)alignment of MUAPs with respect to MEP. The average proportion of accounted energy in LP-filtered MEPs by the identified MU firings in the two highlighted channels was 57% (10% RMT), 49% (30% RMT), 16% (50% RMT), 23% (70% RMT) and 33% (90% RMT). *B*, the firing rate of MUs during a voluntary contraction 1000 ms prior to stimulation. *C*, the probability of a MU firing during MEPs as a function of pre-stimulus voluntary MU firing rate (with 95% confidence intervals). *D*, maximal peak-to-peak (P2P) amplitude of motor evoked potentials (MEPs) across LP-filtered HDsEMG channels. *E*, the number of identified motor unit (MU) firings during MEPs. *F*, the proportion of accounted energy in LP-filtered MEPs by the identified MU firings (energy accounted for, EAF). *G*, the latency of the identified MUs during MEPs. *H*, the standard deviation of latencies of all the identified MU firings during MEPs ($SD_{MULat,MEP}$), indicating dispersion of MU firing identification. For *B* and *D–H*, data are presented as means with SD shown as bars, with individual participant averages shown as circles. The horizontal lines denote the statistical difference between different contraction levels: *$P < 0.05$, **$P < 0.01$, ***$P < 0.001$. [Colour figure can be viewed at wileyonlinelibrary.com]

(Holobar et al., 2014) and accounted for a high proportion of LP-filtered simulated MEP energy. Both precision and sensitivity were influenced by the stimulation intensity, typically being lower at higher stimulation intensities. When considering merely the transfer of MU filters from individual submaximal voluntary isometric contractions, particularly those from lower contraction levels, the effect of stimulation intensity was the largest. This is expected as the proportion of the activated motor pool increases with contraction or stimulation level, resulting in higher number of recruited and detected MUs. Therefore, for simulated MEPs of higher amplitude, low-threshold MUs exhibit much lower energy of MUAPs with respect to other MUs in the surface EMG signal (regarded as physiological noise) and thus exhibit lower physiological signal-to-noise ratio at higher compared to lower contraction/stimulated levels (Francic & Holobar, 2021). The difficulty in identifying low-threshold MUs in large MEPs notwithstanding, stimulation intensities that elicit MEPs corresponding to the recruitment of the entire motor pool are rare in experimental conditions; for example, even maximal MEP amplitudes do not reach the amplitudes equivalent to the maximal M-wave (Brouwer & Ashby, 1990; Peri et al., 2017).

The sensitivity was systematically lower than precision, which suggests that some MUs might be missed when MU filters are applied from voluntary to elicited contractions, especially when MU filters are estimated from lower voluntary contraction levels and applied to MEPs elicited at higher stimulation intensities. This is a common phenomenon observed in decomposition of signals acquired during strong contractions, whereby a greater superimposition of MUAPs will render the identification of lower threshold MUs significantly more difficult in the presence of higher threshold MUs (Francic & Holobar, 2021). However, the false alarm rates (calculated as 1 − precision) were generally found to be low (<5%) and comparable to voluntary contractions of higher force levels (Holobar et al., 2014). Overall, the simulation results provide evidence that the method described herein is accurate and robust for conditions typically expected in experimental settings and supports the firing identification of the relatively large number of active MUs.

Unlike in synthetic signals, the ground truth about MU firings is not known in experimental signals, and thus a different approach was taken to validate the feasibility of the methodology. The gold-standard approach to validate MU identification via HDsEMG has been to compare the agreement of MU firing moments from independent decomposition of simultaneously recorded surface and intramuscular EMG signals (De Luca et al., 2006). However, such an approach suffers from a small number of MUs that are identified by both approaches (Holobar et al., 2010). Therefore, we relied on a series of indirect, signal-based metrics to establish the validity of the methodology.

Firstly, given that the methodology is inherently based on the ability to establish MU filters from voluntary contractions, we assessed the reliability of those MU filters throughout an experimental session. To that end, MU filters were estimated from voluntary contractions performed at the beginning of the experiments and transferred to voluntary contractions performed at the end of the session. We demonstrated that whilst the PNR changed from pre- to post-session contractions, the values were on average above 30 dB, which has been shown to be associated with >90% accuracy and <5% false alarm rate of MU identification (Holobar et al., 2014). These results support the notion of reliability of MU filters, the estimation and application of which allow reliable identification of MU firings during MEPs.

Secondly, we assessed the variability of MU identification, which has been associated with decomposition errors (Holobar et al., 2010; Marateb et al., 2011; Yavuz et al., 2015). Rather than focusing on the variability of the interspike interval (Holobar et al., 2010; Marateb et al., 2011), we calculated the variability of the latency of MU firings during MEPs in the present study (i.e. dispersion of identified MU firings). Note that this variability will be in part influenced by the detection delay as a function of MUAP conduction velocity, as well as dispersion of the innervation zones and end plate locations (Fig.3). The variability in MU firings was shown to be very low in both FDI and TA throughout the MEP recruitment curve, comparable to studies that identified MUs during MEPs with intramuscular EMG (Palmer & Ashby, 1992; Brouwer & Ashby, 1992). The variability of MU firings was slightly greater when eliciting MEPs in TA during voluntary contractions (5.4 (2.0) ms). This is likely because of the relative difficulty in segmenting elicited from voluntary MU firings when both types of activity are superimposed. Nevertheless, the average variability of the firing latencies of the identified MUs during MEPs elicited during voluntary efforts is similar to the upper range of MU firing variability previously shown in the results obtained from intramuscular signals (Brouwer & Ashby, 1992).

Lastly, we assessed the representation of the identified MU firings with respect to the MEP recorded by HDsEMG; this was done by calculating the proportion of energy accounted for by the identified MU firings in MEPs. The energy accounted for by MEP decomposition was not influenced by stimulation intensity in TA (either at rest or during a background contraction), but was in FDI, with the values being lower for stimulations at 110% RMT. This is likely a consequence of the difficulty in identifying the very low threshold MUs that contribute to MEPs evoked with an intensity 10% above the RMT, a limitation inherent to HDsEMG decomposition with blind source

separation algorithms (Farina et al., 2010; Holobar et al., 2010). It is important to note that with HDsEMG decomposition, the residual will never be zero; indeed, the contribution of distant MUs to the multichannel signal cannot be reliably identified (Farina et al., 2010; Holobar et al., 2010, 2014). However, the proportion of the energy accounted for by the identified MU firings in MEP in the present study is comparable to the levels observed for extensively validated voluntary contractions (Holobar et al., 2010, 2014). These results suggest that the ability to identify firings of individual MUs from HDsEMG decomposition is similar for voluntary and elicited contractions.

## Physiological insights

In addition to the signal-based metrics, the physiological insights gained from the results of our study provide further support for the validity of the described methodology. We demonstrated that with increased stimulation intensity that evoked greater MEP amplitude, the number of the identified MUs during MEPs also increase, and these two variables were positively correlated. The narrow peaks of post-stimulus time histograms (Fig.6) are also consistent with the notion that TMS recruits pyramidal neurons in the corticospinal tract with a large monosynaptic component (Nielsen et al., 2003). Furthermore, the number of identified MUs during MEPs was very low for subthreshold stimulations (FDI: 0.2 (0.2) MUs; TA: 1.0 (1.4) MUs at 90% RMT), suggesting an inherent low risk of false positive identification, at least during very low stimulation intensities.

We also explored the MU recruitment order with TMS. The classical calculation of MU recruitment thresholds based on force recordings during MEPs is challenging due to a compressed MU recruitment range as a result of very high recruitment speeds, coupled with an inconsistent ability to record a twitch in a resting muscle (Todd et al., 2016). Furthermore, MU firing delays stemming from differences in MUAP conduction velocities and differences in the location of innervation zones (Fig.3) prevent accurate estimation of recruitment order as a function of time. Instead, we focused on the number of firings of the identified MUs from the same units (i.e. the response probability) across the entire MEP recruitment curve. The hypothesis was that the occurrence of firing of a given MU would be more frequently observed throughout the MEP recruitment curve if that unit was recruited at a lower stimulus intensity. We showed a negative association between the number of firings of the identified MUs across the entire MEP recruitment curve and the voluntary recruitment threshold of the same units. In other words, the MUs with the highest voluntary recruitment threshold were the least commonly identified during MEPs throughout the recruitment curve,

suggesting orderly recruitment of MUs with TMS. This is in agreement with studies that have shown similar behaviour of MUs in response to TMS using intramuscular EMG (Gandevia & Rothwell, 1987; Bawa & Lemon, 1993). However, that previous studies employing intramuscular EMG have reached the conclusion on orderly recruitment with recordings based on testing only the first recruited MU (i.e. the one that had been identified to be recruited first from voluntary contractions; Gandevia & Rothwell, 1987), or a slightly greater number of MUs identified via a large number of stimuli (>100), but with a limited range of stimulation intensities (Bawa & Lemon, 1993). In the present study, we confirmed the results of prior work employing intramuscular EMG and extended the observations by showing the orderly MU recruitment with TMS in a larger population of identified MUs, throughout the full range of stimulation intensities (up to the maximal MEP amplitude in Experiment 3), and with fewer stimuli per stimulation intensity (8–20 stimuli).

With regard to MU latencies during MEPs, these were lower for FDI compared to TA MUs, consistent with the differences in central conduction times. The results on the estimated MU latencies in response to TMS are less clear. The firing latency of individual MUs will depend on a number of factors, including the excitability of pyramidal tract and spinal motoneurons, the strength and duration of the synaptic volley, the conduction velocity of single fibre action potentials, the dispersion of innervation zones and end plate locations, and the relative ability to identify low-threshold MUs across the entire recruitment curve, among others. Thus, comparison of individual MU latencies as a function of stimulus strength either between or within individuals may be challenging. Indeed, previous work employing intramuscular EMG recordings also reported heterogeneous behaviour of MU latencies (Bawa et al., 2002). In our study, FDI MU latencies were greater during MEPs evoked at 140% RMT compared to lower intensity (by 0.1–0.2 ms on average), though this difference may be considered negligible due to a lower sampling rate of HDsEMG recordings, whereby an observable difference in latency is only possible in the order of 0.5 ms. In the subsequent experiments in TA, the sampling rate was thus increased. The latencies of the identified MUs in TA were longer during MEPs evoked at 110% RMT compared to the responses of greater amplitude, possibly in response to evoking a D wave. The latencies of the identified MUs were then even shorter in MEPs elicited with 180% RMT, likely due to the difficulty of identifying firings of lower-threshold MUs during a maximal MEP amplitude.

The TA MU latencies were also slightly shorter when MEPs were elicited during voluntary efforts compared to in a resting muscle. This is a common occurrence that has been demonstrated previously for stimulation of

corticospinal axons (Petersen et al., 2002). The average decrease in latency with background contraction level is likely due to the variability of the excitability of motoneurons between refractoriness and threshold, resulting in an increase in resting membrane potential and thus an increase in composite excitatory postsynaptic potentials of motoneurons (Petersen et al., 2002). In turn, this change in membrane potential will also increase the variability in latency as demonstrated in the present study (2.7 (1.3) *vs.* 4.5 (4.4) ms in TA during rest and voluntary contraction, respectively).

Lastly, we investigated the relationship between the voluntary MU firing rate and the probability of firing of a given MU in response to TMS delivered during a sustained isometric contraction. We employed a logistic binomial regression model and demonstrated that there was an inverse association between the background MU firing rate and the occurrence of a given MU firing, suggesting that the probability of firing of a given MU during MEPs decreased with an increase in voluntary MU firing rate. These results are consistent with prior reports that identified single MU firings during MEPs with intramuscular EMG recordings (Brouwer et al., 1989; Bawa & Lemon, 1993). For stimuli greater than synaptic noise (Matthews, 1999), a decrease in the probability of an evoked response with increased voluntary MU firing rate is likely the result of increased depth and steepness of the asymptotic region of the motoneuron afterhyperpolarisation phase (Jones & Bawa, 1999). Alternatively, or in addition to that, with repetitive motoneuron activation, the dendritic membrane might be depolarised, resulting in decreased responsiveness of a motoneuron to additional synaptic input (Lee & Heckman, 2000; Fuglevand et al., 2015). Thus, the MEP amplitude is likely to increase to a point when most MUs are voluntarily recruited, after which the effect of decreased firing probability will be the principal mechanism modulating the aggregate MEP amplitude (Oya et al., 2008). Indeed, it has been shown that the MEP amplitude increases with increased level of background voluntary contraction to a certain contraction level, after which it either plateaus or decreases depending on the investigated muscle (Oya et al., 2008; Škarabot et al., 2018; Škarabot, Ansdell, Brownstein, Thomas et al., 2019). We found that the number of identified MUs was greater for MEPs evoked during a voluntary contraction at 30% compared to 70 and 90% MVF, without a concomitant decrease in MEP amplitude; this suggests that some MUs had gone undetected likely due to the difficulty in identifying small, lower-threshold MUs in the presence of larger, higher-threshold MUs (Francic & Holobar, 2021). In contrast to our experiment that only used five stimuli at each contraction intensity, studies using intramuscular EMG had to rely on a substantially greater number of evoked responses (e.g. >100 stimuli; Bawa & Lemon, 1993) to reliably quantify the firing probability over time due to a small number of identified MUs per single response. In the present study, the identification of many TA MU firings per individual evoked response (45.3 (11.6) MUs per individual) from HDsEMG facilitated our ability to calculate the firing probability on the MU population level with substantially fewer number of stimuli.

Taken together, we showed that the physiological insight gained from the decomposition of the HDsEMG signals to identify MU firings during MEP is consistent with prior reports using intramuscular EMG, thus providing support for the applicability of the methodology in neurophysiological studies. Notably, these physiological insights were demonstrated by identifying firings of a substantially greater number of MUs during MEPs that are more likely to be representative of the behaviour of the entire motor pool, and by identifying firings of MUs with a substantially greater range of recruitment thresholds (up to ~60% MVF), typically with fewer required stimuli compared to MU identification from intramuscular EMG signals.

## Further considerations and potential applications

Whilst we demonstrated the feasibility of the methodology and showed its accuracy in identifying MU firings constituting MEPs, there are a number of factors to consider when applying the methodology and some limitations that bear further exploration in future studies. Firstly, the principle of the methodology relies on the ability to estimate MU filters during voluntary contractions and their application to contractions elicited by TMS to identify MU firings. This inevitably means that some MUs will not be identified during MEPs. These are likely to be (1) MUs that are further away from the recordings site; (2) MUs with higher thresholds which are recruited during higher levels of voluntary contraction force, during which superimposition of MUAPs impedes one's ability to segment those units reliably from crosstalk (i.e. other simultaneously active MUs; Francic & Holobar, 2021); and (3) MUs with very low recruitment thresholds due to the inherent bias of blind source separation algorithms such as the CKC method to identify larger MUs with larger MUAPs (Farina et al., 2010). This was evidenced by a slightly poorer performance of the proposed method to identify firings of individual MUs for MEPs evoked with very high stimulation intensities, or for MEPs evoked during forceful background contraction levels (>50% MVF). Furthermore, because the identified MUs are limited to those recruited during voluntary efforts, the methodology will likely miss phasic MUs recruited by the stimulation (Bawa & Lemon, 1993).

Secondly, the feasibility of the methodology has been demonstrated in FDI and TA muscles. Because both muscles are distal, the differences in conduction velocity and longer conduction times will result in a greater dispersion of MU firing latencies, which likely facilitated the success of the method. Furthermore, because of the geometry and relatively low levels of adipose tissue, HDsEMG signals recorded from these muscles typically yield higher number of identified MUs compared to some proximal muscles (Del Vecchio et al., 2020). We have previously identified MU firings during evoked reflexes in the soleus, another distal muscle (Kalc et al., 2022a). Thus, the methodological approach presented herein should be evaluated further on other muscle groups, particularly proximal and/or fusiform muscles.

The ability to study responses to TMS on the level of individual MUs in a non-invasive manner offers a range of possibilities that could be addressed in future studies. For example, the study of peaks in the post-stimulus time histograms and their relation to the number of synapses involved in transmitting the inputs from the cortico-spinal tract (Nielsen et al., 2003), and the analysis of the peri-stimulus frequencygram to discern facilitatory and inhibitory inputs to motor pools (Yavuz et al., 2015) require exploration in future studies. The examination of peaks in the post-stimulus time histogram may also offer the possibility to discern the occurrence and peri-odicity of multiple volley response to TMS (Sakai et al., 1997; Terao et al., 2000, 2001), though this has been somewhat debatable as multiple peaks are not necessarily observed in all muscles (Brouwer & Ashby, 1990, 1992), as also demonstrated herein (Fig.7). Notably, and as demonstrated in the present study, a smaller number of stimuli may be required with the method described in the present study to construct reliable histograms of single MU responses compared to intramuscular EMG. Lastly, given that one is able to track the same MUs identified by HDsEMG decomposition, either via MUAP waveform correlation (Martinez-Valdes et al., 2017; Del Vecchio & Farina, 2020) or re-application of MU filters (Francic & Holobar, 2021), the presented methodology may allow a study of the plasticity of corticospinal inputs to the same MUs.

## Conclusion

We described and validated a new methodology for the non-invasive identification of firings of a large population of MUs from HDsEMG in response to single-pulse TMS. By simulating synthetic HDsEMG signals we showed that the methodology, which is based on estimation of MU filters from voluntary contractions and their application to elicited contractions, is accurate in identifying MU firings during complex and highly synchronised responses

to TMS. Through a series of experimental studies, we further demonstrated the feasibility of the method to identify firings of a large number of MUs throughout the full MEP recruitment curve, as well as when stimuli were delivered during voluntary efforts. The described methodology demonstrates an emerging possibility to study responses to TMS on a level of individual MUs in a non-invasive manner.

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

# Additional information

## Data availability statement

Data are available from the corresponding authors upon reasonable request.

## Competing interests

The authors declare they have no conflict of interest.

## Author contributions

J.Š., C.A., G.F. and A.H. conceived and designed the work; J.Š., T.G.B., M.D., F.U., C.A., G.F. and A.H. performed experiments; J.Š., N.M. and A.H. analysed the data; J.Š., G.F. and A.H. interpreted results; J.Š. and A.H. prepared the figures; J.Š. drafted the manuscript; all authors revised manuscript. All authors have read and approved the final version of this manuscript and agree to be accountable for all aspects of the work in ensuring that questions related to the accuracy or integrity of any part of the work are appropriately investigated and resolved. All persons designed as authors qualify for authorship, and all those who qualify for authorship are listed.

## Funding

This study was supported by the Slovenian Research Agency (Project J2-1731 and Programme funding P2-0041, awarded to A.H., M.D. and F.U.). JŠ. is supported by Versus Arthritis Foundation Fellowship (reference: 22 569). M.D., F.U., N.M., and A.H. are supported by Horizon Europe Research and Innovation Programme (No. 101079392). C.A. is supported by Comunidad de Madrid fellowship (2017-T2/BMD-5231) and Juan de la Cierva fellowship (IJC2020-04 5437-I). G.F. is supported by 'la Caixa' Foundation (grant LCF/PR/HR20/52 400 012) and by Ministerio de Ciencia e Innovación, Agencia Estatal de Investigación (Spain) with co-funding by the European Regional Development Fund of the European Union (grant PID2021-128623OB-I00).

## Acknowledgements

The authors are grateful to Mr Ayomide Florian Nurudeen for assistance with data collection.

## Keywords

decomposition, high-density surface electromyography, motor evoked potential

# Supporting information

Additional supporting information can be found online in the Supporting Information section at the end of the HTML view of the article. Supporting information files available:

**Statistical Summary Document**
**Peer Review History**
**Supplementary Material**

