## [Peer Review History · The Journal of Physiology]

Decoding firings of a large population of human motor units from high-density surface electromyogram in response to transcranial magnetic stimulation

Jakob Škarabot, Claudia Ammann, Thomas G Balshaw, Matjaž Divjak, Filip Urh, Nina Murks, Guglielmo Foffani, and Ales Holobar

DOI: 10.1113/JP284043

Corresponding author(s): Jakob Škarabot (J.Skarabot@lboro.ac.uk)

Review Timeline:

Submission Date:	29-Oct-2022
Editorial Decision:	04-Jan-2023
Revision Received:	25-Jan-2023
Editorial Decision:	02-Mar-2023
Revision Received:	09-Mar-2023
Editorial Decision:	13-Mar-2023
Revision Received:	13-Mar-2023
Accepted:	17-Mar-2023

Senior Editor: Richard Carson

Reviewing Editor: Madeleine Lowery

Transaction Report:

Dear Dr Škarabot,

Re: JP-TFP-2022-284043 "Decoding firings of a large population of human motor units from high-density surface electromyogram in response to transcranial magnetic stimulation" by Jakob Škarabot, Claudia Ammann, Thomas G Balshaw, Matjaž Divjak, Filip Urh, Nina Murks, Guglielmo Foffani, and Ales Holobar

Thank you for submitting your manuscript to The Journal of Physiology. It has been assessed by a Reviewing Editor and by 2 expert referees and we are pleased to tell you that it is potentially acceptable for publication following satisfactory major revision.

LANGUAGE EDITING AND SUPPORT FOR PUBLICATION: If you would like help with English language editing, or other article preparation support, Wiley Editing Services offers expert help, including English Language Editing, as well as translation, manuscript formatting, and figure formatting at www.wileyauthors.com/eoo/preparation. You can also find resources for Preparing Your Article for general guidance about writing and preparing your manuscript at www.wileyauthors.com/eoo/prepresources.

REVISION CHECKLIST:

We look forward to receiving your revised submission.

Yours sincerely,

Richard Carson
Senior Editor
The Journal of Physiology

REQUIRED ITEMS FOR REVISION:

-Author photo and profile. First (or joint first) authors are asked to provide a short biography (no more than 100 words for one author or 150 words in total for joint first authors) and a portrait photograph. These should be uploaded and clearly labelled with the revised version of the manuscript. See Information for Authors for further details.

-Please upload separate high-quality figure files via the submission form.

-Please ensure that the Article File you upload is a Word file.

-A Statistical Summary Document, summarising the statistics presented in the manuscript, is required upon revision. It must be on the Journal's template, which can be downloaded from the link in the Statistical Summary Document section here: https://jp.msubmit.net/cgi-bin/main.plex?form_type=display_requirements#statistics

-Papers must comply with the Statistics Policy https://jp.msubmit.net/cgi-bin/main.plex?form_type=display_requirements#statistics

In summary:

- If $n \leq 30$, all data points must be plotted in the figure in a way that reveals their range and distribution. A bar graph with data points overlaid, a box and whisker plot or a violin plot (preferably with data points included) are acceptable formats.
- If $n > 30$, then the entire raw dataset must be made available either as supporting information, or hosted on a not-for-profit repository e.g. FigShare, with access details provided in the manuscript.
- n clearly defined (e.g. x cells from y slices in z animals) in the Methods. Authors should be mindful of pseudoreplication.
- All relevant n values must be clearly stated in the main text, figures and tables, and the Statistical Summary Document (required upon revision)
- The most appropriate summary statistic (e.g. mean or median and standard deviation) must be used. Standard Error of the Mean (SEM) alone is not permitted.
- Exact p values must be stated. Authors must not use 'greater than' or 'less than'. Exact p values must be stated to three significant figures even when 'no statistical significance' is claimed.
- Statistics Summary Document completed appropriately upon revision

-Please include an Abstract Figure file, as well as the figure legend text within the main article file. The Abstract Figure is a piece of artwork designed to give readers an immediate understanding of the research and should summarise the main conclusions. If possible, the image should be easily 'readable' from left to right or top to bottom. It should show the

physiological relevance of the manuscript so readers can assess the importance and content of its findings. Abstract Figures should not merely recapitulate other figures in the manuscript. Please try to keep the diagram as simple as possible and without superfluous information that may distract from the main conclusion(s). Abstract Figures must be provided by authors no later than the revised manuscript stage and should be uploaded as a separate file during online submission labelled as File Type 'Abstract Figure'. Please ensure that you include the figure legend in the main article file. All Abstract Figures should be created using BioRender. Authors should use The Journal's premium BioRender account to export high-resolution images. Details on how to use and access the premium account are included as part of this email.

EDITOR COMMENTS

Reviewing Editor:

This paper presents a method for decomposition of motor unit firings in response to transcranial magnetic stimulation (TMS) by extending well-established methods for decomposition of voluntary high density EMG activity. The approach is likely to be of broad interest. However, both reviewers comment on the length of the manuscript, which makes it difficult to read and interpret the results, and have recommended shortening or restructuring the manuscript. Both have also raised some technical issues which should be addressed. In addition, the novel contribution of the presented motor unit identification method with respect to Kalc et al., 'Motor unit identification in the M waves recorded by high-density electromyogram' IEEE Trans Biomed Eng, (2022) should be explained as the overall approach appears quite similar, though applied here to TMS.

Senior Editor:

It is evident that the content of this submission is likely to be of interest to readers of the Journal. As highlighted in the comments provided by the referees and Reviewing Editor however, a good deal of additional work will be required to enhance the presentation of the work, in order to convey the key findings effectively and succinctly. As also emphasised in other comments, there are also several technical matters on which clarification is necessary.

REFEREE COMMENTS

Referee #1:

This study implemented a motor unit (MU) filtering approach with Convolution Kernel Compensation (CKC) method for MU decomposition during voluntary and TMS elicited muscle contractions. The study used both synthetic EMG signals and experimentally acquired EMG signals. MU Decomposition on electrically evoked muscle contraction with synchronized firing activities has been a challenge. The investigation is comprehensive and would be of interests to readers of the journal.

Major concerns:

It would be helpful to clarify if the MU filtering optimization was performed only on the voluntary data and then applied to the data with motor evoked potentials (MEPs), or the voluntary and MEP data were used together for the optimization. If the later, how did the authors deal with stimulation artifacts? It is unclear how the filter weights can remove the artifacts. The authors may consider reducing the color bar range of Fig 1A, so the range of the weights can be more visible.

The methods (line 257-259) stated that one third of the MEPs were removed to facilitate MU decomposition. Would this be equivalent to having a longer voluntary data segment? It was later stated that this procedure was also performed for the sensitivity and specificity calculation. The main focus of the study was to quantify the accuracy of MEP decomposition. The removal of one third of MEPs needs better justification.

The synthetic data showed that less than 10% of MUs were extracted from the data. But the superimposed MUAPs matched very well with the MEPs, which would require a near complete decomposition? Namely, the residual signal near the MEPs would be near zero. It would be helpful to clarify the inconsistency. Similarly, the energy-accounted-for in the experimental data are 20-30%, but the reconstructed waveform is very similar to the actual MEPs.

It seems that the sensitivity and precision were lower at the low contraction level than at the very high contraction level (Figure 4).

Minor concerns:

The manuscript is very long; the authors may consider move some of the experimental descriptions for the later experiments to the supplemental material.

Due to large variation between subjects, the authors may consider using lines connected with the circles instead of plotting average bars. The average does not mean much in Figure 5, 8, 9.

Referee #2:

The authors have performed both simulations and experimental studies on single MU recognition in voluntary contraction and TMS MEPs.

In general the paper and methodology is of great interest and the study is well performed.

My main concern is the readability of the very extensive paper. As the paper includes both simulations and experimental studies, on multiple muscles, the paper is quite hard to digest. Separating the simulations from the experimental study would further provide some room to better explain the simulation study which is no compressed such that details are missing.

Some remarks with respect to the text:

Introduction In 121-122: Why would this be the next logical step?

methods:

2.3.2. It is confusing reading a protocol that lacks the details. reading further these details are provided but at 2.3.3 I recommend rewrtng these two paragraphs to one.

results:

exp 1. many results and significant levels are in the text making it very hard to read and find the relevant results. It would be better to provided them in a table perhaps as addendum and include only those results that are most important.

Fig4. MEPs are plotted up-side down as clinical neurophysiologist are custom to have negative polarity up.

line 718 and further is a full discussion. Again, this simulation part would benefit in readability if described in a separate manuscript.

3.2 it would have been valuable to know the relation with maximum CMAP as an indication for percentage of MUs that might have been recruited

In 758-760, it is not completely clear what is meant here

In 781, seems more appropriate for discussion

discussion

stability of MU filters is discussed, the filters are stable by definition I would think, the question is if the unchanged MU filters can still be applied, which apparently is the case. please rephrase

In1062: only at low level this might be true but this does not mean this is true for higher MEP amplitudes is it?

END OF COMMENTS

Dear Editors and Referees,

Thank you for taking the time to review our manuscript and providing us with the opportunity to improve our work. We have addressed your concerns and those of the referees which we believe have improved our manuscript. You may find our responses to specific comments below. Please note that the references to line numbers now refer to the revised clean manuscript version.

EDITOR COMMENTS

Reviewing Editor:

This paper presents a method for decomposition of motor unit firings in response to transcranial magnetic stimulation (TMS) by extending well-established methods for decomposition of voluntary high density EMG activity. The approach is likely to be of broad interest. However, both reviewers comment on the length of the manuscript, which makes it difficult to read and interpret the results, and have recommended shortening or restructuring the manuscript. Both have also raised some technical issues which should be addressed. In addition, the novel contribution of the presented motor unit identification method with respect to Kalc et al., 'Motor unit identification in the M waves recorded by high-density electromyogram' IEEE Trans Biomed Eng, (2022) should be explained as the overall approach appears quite similar, though applied here to TMS.

Senior Editor:

It is evident that the content of this submission is likely to be of interest to readers of the Journal. As highlighted in the comments provided by the referees and Reviewing Editor however, a good deal of additional work will be required to enhance the presentation of the work, in order to convey the key findings effectively and succinctly. As also emphasised in other comments, there are also several technical matters on which clarification is necessary.

Thank you for allowing us to make improvements on our manuscript. We have made a great number of revisions to improve presentation of the work as follows. In light of your comments, as well as those of both Referees, we have made attempts to shorten the manuscript and thus present the key findings more succinctly. We have taken on board the suggestion of referees to combine certain sections of methodological approach to the experimental work and reduce the repetitiveness of the description of procedures where applicable. Specifically, we have combined descriptions of protocol and procedures to improve clarity and thus made the description more succinct and effective. This has reduced the length of those sections, and overall, the Methods section. Since this is a manuscript describing a technique, we believe that any substantial further shortening of the Methods section would impede the complete understanding of the work as well as impede the application of the method by the scientific community.

Modifications have also been made to the Results section. Specifically, we have streamlined the reporting on the number of identified motor units and their basic statistics (recruitment thresholds, recruitment threshold range, pulse-to-noise ratio etc.) that were previously reported in text, and now present them in a table (Table 1). Additionally, we have made the reporting of the modelling results more succinct considering suggestions of Referee #2; some reporting of specific results is now

done via the figure and supplementary material. Consequently, the text in the Results section is now limited to about a single page per experiment or less (approximately 3.5 pages of text, excluding figures, compared to 6 pages in the original submission). Some revision of sentences has also been made in the Discussion to improve clarity.

Regarding the publications of Kalc et al., please note that these were not published at the time of submission of this manuscript. Nevertheless, there are unique challenges to identifying MU firings in response to TMS that make the methodology proposed in the manuscript novel. In particular, TMS responses are multi-volley responses and therefore much more complex than responses to percutaneous nerve stimulation. This represents difficulties with segmentation of decomposition results (i.e., identifying the true firing with respect to crosstalk), particularly when responses are evoked during voluntary contractions, something which we cover in the Discussion. We have now addressed the differences in identifying MU firings in different types of evoked responses more explicitly in the Introduction (Lines 125-132): *“Promising results have been shown previously whereby MU firings could be identified in induced reflexes (Yavuz et al., 2015; Kalc et al., 2022a), and a limited number of MU firings were identified in response to maximal percutaneous nerve stimulation (i.e., M wave; Kalc et al., 2022b). However, responses to TMS represent their own challenges, namely the greater complexity of the multi-volley compared to single-volley responses such as H-reflexes and M-waves (Brouwer & Ashby, 1990, 1992; Bawa & Lemon, 1993).”*

Furthermore, in this study, we performed simulations that are more directly relevant to responses to TMS. Namely, we simulated the MEP recruitment curve, which is often used in clinical neurophysiological studies, using physiological MU synchronisation levels relevant to TMS. We have now addressed this in the Discussion (Lines 776-784): *“To assess the feasibility of the methodology we first modelled synthetic HDsEMG signals for which the ground truth about MU firings was known. We previously showed in simulations that the approach of estimating MU filters from the decomposition of voluntary submaximal contractions and applying them to evoked responses allows accurate identification of MU firings at a fixed proportion of motor pool recruitment, even in cases of supraphysiological synchronisation of firings (Kalc et al., 2022b, 2022a). Here, we simulated MEPs corresponding to a wide range of the proportion of motor pool activation (10-100%) and demonstrated high precision and accuracy of MU identification (>90%).”*

Moreover, in the present manuscript we present the application of the methodology on different muscles, which provides comprehensive evidence of feasibility of the methodology. Nevertheless, as we cover in the Discussion, application on other muscles, particularly proximal and/or fusiform muscles might represent unique challenges for the methodology (see Line 1007-1010): *“We have previously identified MU firings during evoked reflexes in the soleus, another distal muscle (Kalc et al., 2022a). Thus, the methodological approach presented herein should be evaluated further on other muscle groups, particularly proximal and/or fusiform muscles.”*

Lastly, in the present study we provide a number of physiological insights that extend the findings previously reported using intramuscular EMG techniques including, orderly recruitment of MUs up to maximal MEP amplitude, and decreased probability of MU firing during responses evoked with greater contraction level (evidence with contractions up to 90% MVF), in a large number of identified MUs with fewer required stimuli, something which we address in the Discussion, for example (Lines 972-980):

“Taken together, we showed that the physiological insight gained from the decomposition of the HDsEMG signals to identify MU firings during MEP is consistent with prior reports using intramuscular EMG, thus providing support for the applicability of the methodology in neurophysiological studies. Notably, these physiological insights were demonstrated by identifying firings of a substantially greater number of MUs during MEPs that are more likely to be representative of the behaviour of the entire motor pool, and by identifying firings of MUs with a substantially greater range of recruitment thresholds (up to ~60% MVF), typically with a fewer number of required stimuli compared to MU identification from intramuscular EMG signals.”

REFEREE COMMENTS

Referee #1:

This study implemented a motor unit (MU) filtering approach with Convolution Kernel Compensation (CKC) method for MU decomposition during voluntary and TMS elicited muscle contractions. The study used both synthetic EMG signals and experimentally acquired EMG signals. MU Decomposition on electrically evoked muscle contraction with synchronized firing activities has been a challenge. The investigation is comprehensive and would be of interests to readers of the journal.

Thank you for taking the time to review our manuscript and providing constructive feedback. We have taken on board your suggestions and clarified our approach where necessary. Please find our responses to your comments below.

Major concerns:

It would be helpful to clarify if the MU filtering optimization was performed only on the voluntary data and then applied to the data with motor evoked potentials (MEPs), or the voluntary and MEP data were used together for the optimization. If the later, how did the authors deal with stimulation artifacts? It is unclear how the filter weights can remove the artifacts. The authors may consider reducing the color bar range of Fig 1A, so the range of the weights can be more visible.

MU filters were optimised only on the voluntary data, and once optimised, were applied to concatenated MEP data. We have now clarified this (Line 481-484): “. MU filters were then applied from voluntary to elicited contractions (Figure 2C; for example, see Figure 2F). *Therefore, MU filters were calculated and optimised only on voluntary signals, with no additional optimisation of MU filters performed once they were applied to elicited contractions.*”

Before voluntary and MEP data were concatenated for the purpose of applying filters from the former to the latter, stimulation artifact blanking was performed with a 5 ms window following the stimulation trigger. This has now been clarified in the Methods section (Line 479-481): “*For elicited contractions, stimulation artifacts were removed using artifact blanking procedure with a 5 ms window from the stimulation trigger.*”

As suggested, we have revised the colour bar range in Fig 1A such that the weights of filters are now more visible. Specifically, we assigned a different colour bar range for

each filter, scaled to the range of minimal and maximal coefficients of filter weights. Note that for MU filter 3, the weights with the higher coefficients are somewhat hidden by layers of lower filter weights. We have now added this explanation into figure caption for clarity.

The methods (line 257-259) stated that one third of the MEPs were removed to facilitate MU decomposition. Would this be equivalent to having a longer voluntary data segment? It was later stated that this procedure was also performed for the sensitivity and specificity calculation. The main focus of the study was to quantify the accuracy of MEP decomposition. The removal of one third of MEPs needs better justification.

We apologise for the lack of clarity. We did not remove one third of the MEPs, but one third of randomly selected MU firings in each MEP. We have now revised the sentence to make this clearer (see Line 250-251): “*For each simulated MEP, we removed one third of randomly selected MU firings.*” As stated in subsequent sentences, this was necessary as simulated MUs would have shared too many firings to support accurate recognition of which MU was identified from simulated MEPs. Therefore, this removal of MU firings is not related to the length of voluntary signals, but was introduced in order to facilitate reliable assessment of MU firings in the case of highly synchronised MU firings (i.e., during simulated MEPs).

The synthetic data showed that less than 10% of MUs were extracted from the data. But the superimposed MUAPs matched very well with the MEPs, which would require a near complete decomposition? Namely, the residual signal near the MEPs would be near zero. It would be helpful to clarify the inconsistency. Similarly, the energy-accounted-for in the experimental data are 20-30%, but the reconstructed waveform is very similar to the actual MEPs.

Thank you for allowing us to clarify. You are correct in suggesting that near-complete decomposition of surface EMG signals (i.e., little to zero residual) is practically impossible due to many MUs being too small or being located too far from the recording area to be detected, or due to their relatively small contribution in terms of energy. This limitation is shared for both voluntary and elicited contractions (see ‘Further considerations’ section in the Discussion).

Our results of EAF are consistent with previous reports on decomposition of monopolar voluntary signals where decomposition typically accounts for ~10-30% (e.g., Holobar et al. 2010, 2014). Here, we demonstrate that EAF values for elicited contractions are similar to those during voluntary contractions. However, please note that the energy accounted for is a measure of the accounted energy across the entire grid of electrode, as is stated in Equation 7. Depending on the channel, the accounted energy may therefore be greater/smaller than the average of the collective number of channels recorded by the grid electrode (due to, for example, differences in distribution of motor unit territories). Furthermore, even a small (e.g. half of an intersample interval) misalignment between the MUAPs as estimated by our methodology and the recorded MEP can result in rather large residual after subtraction (please see Equation 7) and thus lower EAF value; this misalignment might indeed vary across the channels of the grid within individual, as well as between-individuals. We have attempted to correct the potential misalignment by manually subtracting the conduction delays (see Methods), though ensuring perfect alignment is practically impossible. Indeed, the EAF values do vary between individuals as demonstrated in our data (e.g. in the FDI muscle in Fig 5D one observes EAF values to vary between approximately 15-45% between individuals). We have now

added clarification in figure captions (where relevant) that for the sake of clarity, only two channels are displayed, but that the representatives of waveform might vary across the channels due to the aforementioned factors: *“For clarity, only two channels are shown, but note that responses and thus the proportion of the reconstructed waveform with respect to the recorded MEPs may vary across the channels due to factors such as distribution of MU territories and (mis)alignment of MUAPs with respect to MEP.”*

It seems that the sensitivity and precision were lower at the low contraction level than at the very high contraction level (Figure 4).

This was indeed the case, and is likely the result of the fact that MU filters obtained from less complex lower contraction intensities are less successful in identifying MU firings during higher amplitude MEPs (or higher voluntary contraction intensities for that matter, see Francic & Holobar 2021 IEEE Access). Note that despite lower precision and sensitivity values, fewer units are identified when applying filters acquired from higher contraction levels (see number of identified MUs displayed in bars in Figure 4; also, a note is made of this in the caption). We have now clarified that in the Results section (see Line 630-634): *“Though fewer MUs were identified at higher than at lower contraction levels, both precision ($\chi^2(4) = 327.3, p < 0.0001$) and sensitivity ($\chi^2(4) = 209.9, p < 0.0001$) of MU firing identification were greater for MU filters acquired during higher compared to lower contraction levels (Figure 4 and C).”*

We have now also clarified this phenomenon in the Discussion (see line 788-801): *“When considering merely the transfer of MU filters from individual submaximal voluntary isometric contractions, particularly those from lower contraction levels, the effect of stimulation intensity was the largest. This is expected as the proportion of the activated motor pool increases with contraction or stimulation level, resulting in higher number of recruited and detected MUs. Therefore, for simulated MEPs of higher amplitude, low-threshold MUs exhibit much lower energy of MUAPs with respect to other MUs in the surface EMG signal (regarded as physiological noise) and thus exhibit lower physiological signal-to-noise ratio at higher compared to lower contraction/stimulation levels (Francic & Holobar 2021). The difficulty in identifying low-threshold MUs in large MEPs notwithstanding, stimulation intensities that elicit MEPs corresponding to the recruitment of the entire motor pool are rare in experimental conditions; for example, even maximal MEP amplitudes do not reach the amplitudes equivalent to the maximal M-wave (Brouwer & Ashby, 1990; Peri et al., 2017).”*

Minor concerns:

The manuscript is very long; the authors may consider move some of the experimental descriptions for the later experiments to the supplemental material.

We have made several modifications to the manuscript in order to make it clearer and more succinct. These are detailed in our response to the editors (see above). In line with journal instructions to authors we are not able to move experimental descriptions into the supplementary material (*“Materials such as figures, tables, text (e.g. expanded/detailed methods or results,... if essential for the complete understanding of the manuscript and can fit on a printed page or pages, must be incorporated into the manuscript*

itself as part of the text... and not supplied as supporting information.”). **However, as suggested by Referee #2, we have combined experimental protocol and procedure sections, reduced the repetitiveness of the details of certain procedures where possible, and thus made the experimental descriptions more succinct. We hope that you will find these satisfactory.**

Due to large variation between subjects, the authors may consider using lines connected with the circles instead of plotting average bars. The average does not mean much in Figure 5, 8, 9.

Thank you for pointing this out. In addition to the average bars (which we believe have some value in terms of interpreting the results of statistical tests for the reader) and circles for individual responses, we have now added the connecting lines between circles to better represent outcome variables in relation to RMT/contraction intensity in figures 5, 8 and 9, as suggested.

Referee #2:

The authors have performed both simulations and experimental studies on single MU recognition in voluntary contraction and TMS MEPs.

In general the paper and methodology is of great interest and the study is well performed.

My main concern is the readability of the very extensive paper. As the paper includes both simulations and experimental studies, on multiple muscles, the paper is quite hard to digest. Separating the simulations from the experimental study would further provide some room to better explain the simulation study which is no compressed such that details are missing.

Thank you for taking the time to review our manuscript and providing constructive comments. Regarding the inclusion of both simulated and experimental data in one manuscript, we believe that these makes our work more robust and comprehensive. Often, when simulations using synthetic EMG signals are presented on their own, the question becomes whether these principles can be applied to experimental signals where conditions are different compared to synthetic signals (e.g., signal-to-noise ratios may vary among recordings, adipose tissue differences between participants and/or application to muscles with different geometries might affect decomposition yield, accuracy of MU identification and therefore the transfer of filters from voluntary to evoked contractions etc.). We believe that presenting the simulation results first to establish the proof-of-concept (and the theoretical and conceptual underpinnings of the methodology), followed by the application to experimental signals comprehensively shows the validity and feasibility of the proposed methodology. Furthermore, the experimental signals were recorded in different muscles (with different geometries which likely affect decomposition) and different conditions (resting, superimposed on voluntary contractions – represent different methodological challenges in terms of segmentation) which further provide very comprehensive evidence of the methodological feasibility. Lastly, the combination of synthetic and experimental signals together showcases the validated utility of the proposed methodology to gain physiological insights.

At the end of the Introduction, we now clarify that the use of synthetic signals was important to first establish the precision and sensitivity of the proposed methodological approach, whereas the use of experimental signals allowed us to establish its feasibility (see Line 137-143): ***“Using synthetic signals with known MU firings we demonstrate the high precision and sensitivity of the methodological approach. Thereafter, using experimental HDsEMG signals obtained from two human muscles with strong corticomotoneuronal projections (first dorsal interosseus, FDI; and tibialis anterior, TA; Palmer & Ashby, 1992; Brouwer & Ashby, 1992), we show the feasibility of decoding firings of a large population of MUs in response to TMS of the human motor cortex in a resting and voluntarily contracting muscle.”***

You will also note that in response to one of your suggestions below, we have streamlined the reporting of the simulation results with some specific details of the statistics now located in the figure and Supplementary Material. Hopefully, these changes now make this section clearer and easier to follow.

Some remarks with respect to the text:

Introduction In 121-122: Why would this be the next logical step?

Thank you for pointing this out. We agree with the reviewer that the sentence was phrased ambiguously and have therefore revised the section to (see lines 119-125): ***“The use of this methodology has greatly advanced our understanding of motor control in health (Farina et al., 2016) and disease (Gallego et al., 2015a, 2015b; Puttaraksa et al., 2019). Nevertheless, HDsEMG decomposition has largely been limited to voluntary contractions, whereas less is known about MU firings underpinning elicited contractions, such as in response to TMS. The nature of elicited contractions represents a computing challenge due to highly synchronised and superimposed MU firings.”***

Methods:

2.3.2. It is confusing reading a protocol that lacks the details. reading further these details are provided but at 2.3.3 I recommend rewriting these two paragraphs to one.

As suggested, we have now combined the sections/rewritten them into one. For consistency, we have similarly combined sections pertaining to the protocol and procedures for Experiments 3 and 4.

results:

exp 1. many results and significant levels are in the text making it very hard to read and find the relevant results. It would be better to provide them in a table perhaps as addendum and include only those results that are most important.

We now only provide detailed results of the precision and sensitivity analysis where all MU filters are considered together, such as was already the case for PNR results. The statistical results of these analyses that represent the main message are reported in text due to transparency of statistical reporting, in line with the journal guidelines.

Given that the results/main message were similar when considering MU filters acquired from individual contraction levels, we now explain in the text that the results were similar and refer to the figure and supplementary material where more specific, detailed statistical differences may be observed. In light of your comment below, we have now also removed the paragraph that read as a discussion and integrated its information into the Discussion. As such, the results of Experiment 1 are now constrained to a single page with a much more streamlined message. We hope that you will find this satisfactory.

Fig4. MEPs are plotted up-side down as clinical neurophysiologist are custom to have negative polarity up.

We have now inverted the signals such that the polarity is more representative of typical experimental signals.

line 718 and further is a full discussion. Again, this simulation part would benefit in readability if described in a separate manuscript.

Thank you for pointing this out. We have now removed this paragraph and integrated its information into the Discussion. Please see our response above for our argument as to why we believe presenting simulated and experimental data increases the robustness of the findings.

3.2 it would have been valuable to know the relation with maximum CMAP as an indication for percentage of MUs that might have been recruited

We appreciate that concurrent recordings of maximum CMAPs could have been valuable. Unfortunately, we did not record percutaneous nerve stimulations in our experiment so we cannot provide those data. Nevertheless, we report peak-to-peak MEP amplitudes from the surface EMG that shows the behaviour consistent with the literature that reports responses to single pulse TMS in FDI and TA muscles. Furthermore, energy-accounted-for represents a signal-based metric that shows approximately 30% of the recorded signal may be accounted for by the identified MUs.

In 758-760, it is not completely clear what is meant here

These results show a direct relationship between the number of identified MU firings per MEP and the peak-to-peak amplitude of MEP. We have now revised the sentence to make that clearer (Line 655-657): “A positive association was demonstrated between the number of identified MU firings and MEP amplitude ($r_m = 0.65 [0.62, 0.69]$, $p < 0.0001$), meaning that with greater MEP amplitude, more MU firings were identified per MEP.”

In 781, seems more appropriate for discussion

We are slightly unclear about the section the reviewer is referring to as Line 781 in the original manuscript (now Line 661) includes the report of a statistical result. We have,

however, reviewed the paragraph and removed part of the sentence that would have been better served as part of the discussion (“... *possibly due to inability to identify several low threshold MUs contributing to a small MEP during stimulation at 110% RMT*”). Additionally, in the next paragraph (commencing at Line 791 in the original manuscript, now Line 669) we removed a part of the sentence that would have been more appropriately placed in the Discussion (“... *showing a clear and sharp single peak, consistent with data obtained from intramuscular recordings (Palmer & Ashby, 1992)*”). These removals have now been incorporated into the Discussion. We hope that you will find these satisfactory.

Discussion

stability of MU filters is discussed, the filters are stable by definition I would think, the question is if the unchanged MU filters can still be applied, which apparently is the case. please rephrase

We agree that stability may not be the most appropriate term. We have now replaced it with the term ‘reliability of MU filters’, which we believe more appropriately reflects the notion we are trying to relay. We have made these changes throughout the manuscript.

In1062: only at low level this might be true but this does not mean this is true for higher MEP amplitudes is it?

You are correct that we cannot necessarily infer the risk of false positive identification during higher MEP amplitudes from the results of subthreshold stimulation intensities. Our interpretation merely referred to the fact that the method inherently does not have a high risk of false positive identification (i.e. if it did, the number of identified MU firings would have likely been much greater in response to subthreshold stimulation where a great number of MUs is not expected to be recruited). We have now revised the sentence to hopefully make that clearer (Line 878-879): “... *suggesting an inherent low risk of false positive identification, at least during very low stimulation intensities.*”

Dear Dr Škarabot,

Re: JP-TFP-2023-284043R1 "Decoding firings of a large population of human motor units from high-density surface electromyogram in response to transcranial magnetic stimulation" by Jakob Škarabot, Claudia Ammann, Thomas G Balshaw, Matjaž Divjak, Filip Urh, Nina Murks, Guglielmo Foffani, and Ales Holobar

Thank you for submitting your revised Technique to The Journal of Physiology. It has been assessed by the original Reviewing Editor and Referees and has been well received. Some final revisions have been requested.

REVISION CHECKLIST:

We look forward to receiving your revised submission.

Yours sincerely,

Richard Carson
Senior Editor
The Journal of Physiology

REQUIRED ITEMS:

-Papers must comply with the Statistics Policy https://jp.msubmit.net/cgi-bin/main.plex?form_type=display_requirements#statistics

In summary:

-If n {less than or equal to} 30, all data points must be plotted in the figure in a way that reveals their range and distribution. A bar graph with data points overlaid, a box and whisker plot or a violin plot (preferably with data points included) are acceptable formats.

-If $n > 30$, then the entire raw dataset must be made available either as supporting information, or hosted on a not-for-profit repository e.g. FigShare, with access details provided in the manuscript.

-' n ' clearly defined (e.g. x cells from y slices in z animals) in the Methods. Authors should be mindful of pseudoreplication.

-All relevant ' n ' values must be clearly stated in the main text, figures and tables, and the Statistical Summary Document (required upon revision)

-The most appropriate summary statistic (e.g. mean or median and standard deviation) must be used. Standard Error of the Mean (SEM) alone is not permitted.

-Exact p values must be stated. Authors must not use 'greater than' or 'less than'. Exact p values must be stated to three significant figures even when 'no statistical significance' is claimed.

-Statistics Summary Document completed appropriately upon revision

EDITOR COMMENTS

Reviewing Editor:

Reviewer 1 has a small number of remaining comments/suggestions to improve the manuscript. Please consider also the section title 'Modelling on synthetic signals'. The title 'Simulation of synthetic signals' would be more correct.

REFEREE COMMENTS

Referee #1:

The revised manuscript has improved substantially. I only have a few minor concerns that the authors may consider addressing.

The current abstract and introduction still reads like a new decomposition algorithm was developed in this study, rather than an evaluation of an existing algorithm on a new application, which is still novel.

As a follow up question on the reconstructed MEP, instead of only selecting some of the best channels to display, the authors may consider including representative channels.

Referee #2:

The paper is still very long but has improved. I still would be in favour of separating the simulation and experimental results.

Other than that I have no further comments.

END OF COMMENTS

1st Confidential Review

25-Jan-2023

Dear Editors and Referees,

Thank you for providing additional comments for us to consider and further improve our work. We have implemented the suggested changes which we describe in detail below.

EDITOR COMMENTS

Reviewing Editor:

Reviewer 1 has a small number of remaining comments/suggestions to improve the manuscript. Please consider also the section title 'Modelling on synthetic signals'. The title 'Simulation of synthetic signals' would be more correct.

We revised the section title according to your suggestion. For consistency, we have revised all references to 'modelling' with 'simulation' in the manuscript when referring to simulated signals.

REFEREE COMMENTS

Referee #1:

The revised manuscript has improved substantially. I only have a few minor concerns that the authors may consider addressing.

Thank you for providing additional suggestions to improve our paper. We have implemented your suggestions, the details of which we describe below.

The current abstract and introduction still reads like a new decomposition algorithm was developed in this study, rather than an evaluation of an existing algorithm on a new application, which is still novel.

We have made revisions that clarify that the most novel part of the studies described in the manuscript is the application of the methodology to motor evoked potentials.

In the abstract, the first sentence has been revised to: “*We describe a novel application of methodology for high-density surface electromyography (HDsEMG) decomposition to identify motor unit (MU) firings in response to transcranial magnetic stimulation (TMS)*”.

We have added an additional sentence in the Introduction that clarifies that the basic principles of the methodology have been established, but this has been adapted and applied to the more complex motor evoked potentials (see Lines 135-138):

“In this study, we aimed to ascertain the validity and accuracy of identifying firings of a large population of individual MUs that constitute an evoked motor response to single pulse TMS. We utilised the principles of Convolution Kernel Composition (CKC) method of HDsEMG decomposition that we recently developed for identification of MU firings underpinning H-reflex (Kalc et al., 2022a) and M-wave (Kalc et al., 2022b); here, we adapted and applied this methodology to more complex evoked responses to TMS.”

As a follow up question on the reconstructed MEP, instead of only selecting some of the best channels to display, the authors may consider including representative channels.

In the interest of transparency, we have now amended the Figure 5, 8 and 9 to include a display of all LP-filtered channels. This will allow the reader to appreciate the level of channel-to-channel variability in the proportion of the reconstructed waveform with respect to the recorded MEPs (as would also be the case in voluntary contractions).

Regarding the selected two channels that we highlight, the closer inspection suggests that they are relatively representative of the energy accounted for (EAF) by the identified MUs in MEPs. To provide evidence of this, we now report the mean EAF values of the two highlighted channels in the respective figure captions. Specifically, for Figure 5 the average EAF values of the two channels were 20% (110% RMT), 33% (120% RMT), 48% (130% RMT) and 49% (140% RMT). For Figure 8, the average EAF values of the two highlighted channels were 46% (100% RMT), 52% (120% RMT), 32% (140%), 38% (160% RMT), 63% (180% RMT). For Figure 9, the average EAF values of the two highlighted channels were 57% (10% MVF), 49% (30% MVF), 16% (50% MVF), 23% (70% MVF), 33% (90% MVF).

The example of this inclusion is provided for Figure 5 below:

“Figure 5. Identification of firings of a population of motor units (MUs) in first dorsal interosseous (FDI) during the motor evoked potential (MEP) recruitment curve. A: Examples of typical MEPs in FDI in response to transcranial magnetic stimulation (TMS) at 110-140% of resting motor threshold (RMT). High density surface electromyography (HDsEMG) channels (green line) filtered with a Laplacian spatial filter (LP) are displayed along with their sum of spike-trigger averaged MU action potentials (blue line). Two channels are highlighted along with the corresponding identified MU firings (coloured). Note that responses and thus the proportion of the reconstructed waveform with respect to the recorded MEPs may vary across the channels due to factors such as distribution of MU territories and (mis)alignment of MUAPs with respect to MEP. The average proportion of accounted energy in LP-filtered MEPs by the identified MU firings in the two highlighted channels was 20% (110% RMT), 33% (120% RMT), 48% (130% RMT) and 49% (140% RMT).”

Note that there is some additional variability between responses across stimulation/contraction levels. That is, for the same EMG channel, EAF varies across stimulation/contraction levels. This makes it challenging to highlight a pair of channels that are perfectly representative of the EAF values calculated across channels (i.e., the values used in statistical analysis).

We believe that displaying an example of the entire grid of LP filtered channels, along with the highlighted channels (which allow a closer inspection) and the precise EAF values in the captions will allow the readers to fully appreciate the variability in the reconstructed waveform.

Referee #2:

The paper is still very long but has improved. I still would be in favour of separating the simulation and experimental results.

Other than that I have no further comments.

Thank you for recognising our efforts in improving the paper.

Dear Dr Škarabot,

Re: JP-TFP-2023-284043R2 "Decoding firings of a large population of human motor units from high-density surface electromyogram in response to transcranial magnetic stimulation" by Jakob Škarabot, Claudia Ammann, Thomas G Balshaw, Matjaž Divjak, Filip Urh, Nina Murks, Guglielmo Foffani, and Ales Holobar

Thank you for submitting your revised Technique to The Journal of Physiology. It has been assessed by the original Reviewing Editor and has been well received. Some final revisions have been requested.

Please address all the points raised and incorporate all requested revisions or explain in your Response to Referees why a change has not been made. We hope you will find the comments helpful and that you will be able to return your revised manuscript within 2 weeks. If you require longer than this, please contact journal staff: jp@physoc.org.

REVISION CHECKLIST:

We look forward to receiving your revised submission.

Yours sincerely,

Richard Carson
Senior Editor
The Journal of Physiology

REQUIRED ITEMS:

-Papers must comply with the Statistics Policy https://jp.msubmit.net/cgi-bin/main.plex?form_type=display_requirements#statistics

In summary:

-If n {less than or equal to} 30, all data points must be plotted in the figure in a way that reveals their range and distribution. A bar graph with data points overlaid, a box and whisker plot or a violin plot (preferably with data points included) are acceptable formats.

-If $n > 30$, then the entire raw dataset must be made available either as supporting information, or hosted on a not-for-profit repository e.g. FigShare, with access details provided in the manuscript.

-'n' clearly defined (e.g. x cells from y slices in z animals) in the Methods. Authors should be mindful of pseudoreplication.

-All relevant 'n' values must be clearly stated in the main text, figures and tables, and the Statistical Summary Document (required upon revision)

-The most appropriate summary statistic (e.g. mean or median and standard deviation) must be used. Standard Error of the Mean (SEM) alone is not permitted.

-Exact p values must be stated. Authors must not use 'greater than' or 'less than'. Exact p values must be stated to three significant figures even when 'no statistical significance' is claimed.

-Statistics Summary Document completed appropriately upon revision

EDITOR COMMENTS

Reviewing Editor:

No further comments.

Senior Editor:

If the statistical summary document has errors please describe what is incorrect:

Please provide additional information in the statistical summary document. Further descriptive data should be provided. It would also be useful to describe the statistical tests in greater detail.

END OF COMMENTS

2nd Confidential Review

09-Mar-2023

EDITOR COMMENTS

Reviewing Editor:

No further comments.

Senior Editor:

If the statistical summary document has errors please describe what is incorrect: Please provide additional information in the statistical summary document. Further descriptive data should be provided. It would also be useful to describe the statistical tests in greater detail.

Dear Senior Editor,

We have revised the statistical summary document; it now includes the equations that describe our statistical modelling, the descriptive statistics for every variable, as well as the p-values for every post hoc test that was performed. We hope that you will find these satisfactory.

Dear Dr Škarabot,

Re: JP-TFP-2023-284043R3 "Decoding firings of a large population of human motor units from high-density surface electromyogram in response to transcranial magnetic stimulation" by Jakob Škarabot, Claudia Ammann, Thomas G Balshaw, Matjaž Divjak, Filip Urh, Nina Murks, Guglielmo Foffani, and Ales Holobar

We are pleased to tell you that your paper has been accepted for publication in The Journal of Physiology.

Authors should note that it is too late at this point to offer corrections prior to proofing. The accepted version will be published online, ahead of the copy edited and typeset version being made available. Major corrections at proof stage, such as changes to figures, will be referred to the Editors for approval before they can be incorporated. Only minor changes, such as to style and consistency, should be made at proof stage. Changes that need to be made after proof stage will usually require a formal correction notice.

All queries at proof stage should be sent to: TJP@wiley.com

Yours sincerely,

Richard Carson
Senior Editor
The Journal of Physiology

P.S. - You can help your research get the attention it deserves! Check out Wiley's free Promotion Guide for best-practice recommendations for promoting your work at www.wileyauthors.com/eoo/guide. You can learn more about Wiley Editing Services which offers professional video, design, and writing services to create shareable video abstracts, infographics, conference posters, lay summaries, and research news stories for your research at www.wileyauthors.com/eoo/promotion.

IMPORTANT NOTICE ABOUT OPEN ACCESS: To assist authors whose funding agencies mandate public access to published research findings sooner than 12 months after publication The Journal of Physiology allows authors to pay an Open Access (OA) fee to have their papers made freely available immediately on publication.

You can check if your funder or institution has a Wiley Open Access Account here: <https://authorservices.wiley.com/author-resources/Journal-Authors/licensing-and-open-access/open-access/author-compliance-tool.html>